EMBO
Molecular Medicine

# A HIF–LIMD1 negative feedback mechanism mitigates the pro-tumorigenic effects of hypoxia

Daniel E Foxler[1,†] ⓘ, Katherine S Bridge[1,†] ⓘ, John G Foster[1] ⓘ, Paul Grevitt[1] ⓘ, Sean Curry[2], Kunal M Shah[1] ⓘ, Kathryn M Davidson[1] ⓘ, Ai Nagano[1], Emanuela Gadaleta[1] ⓘ, Hefin I Rhys[3] ⓘ, Paul T Kennedy[1] ⓘ, Miguel A Hermida[1] ⓘ, Ting-Yu Chang[4] ⓘ, Peter E Shaw[2], Louise E Reynolds[5] ⓘ, Tristan R McKay[6] ⓘ, Hsei-Wei Wang[4], Paulo S Ribeiro[5] ⓘ, Michael J Plevin[7] ⓘ, Dimitris Lagos[8] ⓘ, Nicholas R Lemoine[1] ⓘ, Prabhakar Rajan[1] ⓘ, Trevor A Graham[5] ⓘ, Claude Chelala[1] ⓘ, Kairbaan M Hodivala-Dilke[5] ⓘ, Ian Spendlove[2] ⓘ & Tyson V Sharp[1,*] ⓘ

## Abstract

The adaptive cellular response to low oxygen tensions is mediated by the hypoxia-inducible factors (HIFs), a family of heterodimeric transcription factors composed of HIF-α and HIF-β subunits. Prolonged HIF expression is a key contributor to cellular transformation, tumorigenesis and metastasis. As such, HIF degradation under hypoxic conditions is an essential homeostatic and tumour-suppressive mechanism. LIMD1 complexes with PHD2 and VHL in physiological oxygen levels (normoxia) to facilitate proteasomal degradation of the HIF-α subunit. Here, we identify *LIMD1* as a HIF-1 target gene, which mediates a previously uncharacterised, negative regulatory feedback mechanism for hypoxic HIF-α degradation by modulating PHD2-LIMD1-VHL complex formation. Hypoxic induction of *LIMD1* expression results in increased HIF-α protein degradation, inhibiting HIF-1 target gene expression, tumour growth and vascularisation. Furthermore, we report that copy number variation at the *LIMD1* locus occurs in 47.1% of lung adenocarcinoma patients, correlates with enhanced expression of a HIF target gene signature and is a negative prognostic indicator. Taken together, our data open a new field of research into the aetiology, diagnosis and prognosis of *LIMD1*-negative lung cancers.

**Keywords** adaptive hypoxic response; HIF-1; LIMD1; lung cancer; tumour suppressor
**Subject Categories** Cancer; Vascular Biology & Angiogenesis

## Introduction

The HIF family of transcription factors are heterodimeric proteins formed of a HIF-α and HIF-β subunit (Wang *et al*, 1995). HIF-α is regulated by intracellular oxygen levels; at physiological oxygen tension (normoxia), two highly conserved proline residues within the oxygen-dependent degradation domain of the HIF-α subunit (P402/564 on HIF-1α; P405/531 on HIF-2α) are hydroxylated by prolyl hydroxylase domain (PHD) proteins. Hydroxylated HIF-α is then recognised and ubiquitinated by the von Hippel–Lindau (VHL) E3 ubiquitin ligase complex, resulting in its degradation by the 26S proteasome (Salceda & Caro, 1997; Maxwell *et al*, 1999; Jaakkola *et al*, 2001; Foxler *et al*, 2012). Under low oxygen (hypoxic) conditions, the hydroxylase activity of the PHD enzymes is inhibited; HIF therefore escapes hydroxylation and degradation to initiate a transcriptional programme of cellular response and adaptation to hypoxia.

Under conditions of chronic hypoxia, a negative regulatory feedback loop is initiated whereby free oxygen from inhibited mitochondrial respiration leads to overactivation of PHDs, causing HIF-α degradation and a desensitised hypoxic response (Ginouves *et al*, 2008). However, neoplastic cells survive under conditions of chronic tumour hypoxia by inhibiting the degradation of HIF (Bertout *et al*, 2008). This is exemplified by *VHL* mutations in clear cell renal carcinomas, leading to sustained HIF-α expression and activity (Rechsteiner *et al*, 2011). In non-small-cell lung cancer (NSCLC), deregulation of the HIF negative feedback loop is far less characterised, even though HIF-α protein expression is implicated as a poor prognostic indicator (Giatromanolaki *et al*, 2001; Kim *et al*, 2005).

1   Centre for Molecular Oncology, Barts Cancer Institute, Queen Mary University of London, London, UK
2   Faculty of Medicine and Life Sciences, University of Nottingham, Nottingham, UK
3   The Francis Crick Institute, London, UK
4   Institute of Microbiology and Immunology, National Yang Ming University, Taipei City, Taiwan
5   Centre for Tumour Biology, Barts Cancer Institute, Queen Mary University of London, London, UK
6   School of Healthcare Science, Manchester Metropolitan University, Manchester, UK
7   Department of Biology, University of York, York, UK
8   Centre for Immunology and Infection, Hull York Medical School and Department of Biology, University of York, York, UK
    *Corresponding author. Tel: +44 (0)20 7882 3848; E-mail: t.sharp@qmul.ac.uk
    †These authors contributed equally to this work

The lung tumour suppressor protein LIMD1 is a member of the Zyxin family of adaptor proteins, initially characterised as signal transducers (Kadrmas & Beckerle, 2004) shuttling between the cytoplasm and nucleus. LIMD1 loss has been identified in lung, breast, head and neck squamous cell carcinomas, and adult acute leukaemia (Sharp *et al*, 2004, 2008; Spendlove *et al*, 2008; Ghosh *et al*, 2010b; Liao *et al*, 2015), and its decreased expression in diffuse large B-cell lymphoma has clinical significance to patient prognosis and disease classification/stratification (Xu *et al*, 2015). *Limd1*-knockout mice have increased lung tumour numbers and volume and decreased survival rate compared to *Limd1*-expressing control mice when either challenged with a chemical carcinogen or cross-bred with *Kras*^G12D mice (Sharp *et al*, 2008) validating its critical role in normal cellular homeostasis. Furthermore, it has been reported that silencing of LIMD1 in multidrug-resistant colorectal carcinoma cells increased their chemosensitivity *in vitro* (Chen *et al*, 2014).

As a scaffold protein, LIMD1 exerts multiple tumour-suppressive functions depending on its binding partners. Basal *LIMD1* gene expression is under the control of PU.1, a member of the Ets family of transcription factors (Foxler *et al*, 2011). LIMD1 can repress cell cycle progression through pRb-dependent and pRb-independent inhibition of E2F (Sharp *et al*, 2004) and regulates Hippo signalling by binding to LATS, causing sequestration of the Hippo kinase complex (Das Thakur *et al*, 2010; Codelia *et al*, 2014; Jagannathan *et al*, 2016). LIMD1 is also part of the Slug/Snail complex that regulates E-cadherin transcription (Ayyanathan *et al*, 2007; Langer *et al*, 2008) in addition to facilitating centrosomal localisation of BRCA2 to prevent aberrant cellular proliferation (Hou *et al*, 2016). Our recent work has shown that LIMD1 is a critical effector of microRNA (miRNA)-mediated gene silencing, a process generally considered to be a global tumour-suppressive mechanism (James *et al*, 2010; Bridge *et al*, 2017).

LIMD1 forms complexes with PHD2 and VHL to post-translationally repress HIF-1α protein levels and therefore HIF-1α-mediated gene activation (Foxler *et al*, 2012; Zhang *et al*, 2015). However, the patho-physiological link between this mechanistic role of LIMD1 within the PHD-LIMD1-VHL HIF regulatory complex and cancer is unknown. Here, we report that *LIMD1* expression is upregulated in hypoxia, through a functional HIF-1α-specific hypoxic response element (HRE) within the CpG island in its promoter. LIMD1 facilitates HIF-1α protein degradation under hypoxic conditions by maintaining the PHD2/VHL/HIF-1α degradation complex, thereby reducing HIF-1α-driven gene activation. Utilising an RNAi-mediated knockdown-rescue system, we have identified that inhibition of hypoxia-driven increase in LIMD1 expression causes HIF-1α protein stabilisation and HIF target gene activation. *In vivo*, inhibition of hypoxia-driven LIMD1 expression results in larger and more vascularised xenograft tumours. Finally, our data provide a molecular mechanistic insight into clinico-pathological data indicating that *LIMD1* loss or haplo-insufficiency correlated with elevated HIF-1α-driven gene expression within lung tumours is associated with poorer patient prognosis.

## Results

### LIMD1 is a HIF-1-responsive gene

Homeostatic signalling pathways often have in-built self-regulatory feedback mechanisms to attenuate their activation (Yosef & Regev,

2011). With this in mind, we hypothesised that *LIMD1* might be a HIF target gene as well as a component of the degradation complex. We therefore assessed endogenous *LIMD1* expression in a panel of cell lines exposed to 1% $O_2$ (henceforth referred to as hypoxia), including transformed/immortalised lines (A549, HeLa, HEK293 and U2OS), non-transformed small airway epithelial cells (SAEC) and primary human dermal fibroblasts (HDF). We observed an increase in *LIMD1* mRNA and protein expression in all cell lines in hypoxia when compared to atmospheric oxygen (20% $O_2$, herein referred to as normoxia) using PHD2 as a positive control and PHD1 as a hypoxia non-responsive gene (Figs 1A–C, and EV1A and B; Stiehl *et al*, 2006).

*In silico* analysis of the *LIMD1* promoter identified three putative hypoxic response elements (HRE 1–3; Fig EV1H; Foxler *et al*, 2011). To assess their functionality, we used a *LIMD1* promoter-driven luciferase reporter construct, spanning 1990-bp upstream from the *LIMD1* transcriptional start site [as predicted by the RefSeq NM_014240.2, which corresponds to nucleotides 45634323-6323 on the primary chromosome 3 ref assembly NC_000003.11 (Foxler *et al*, 2011)] and encompassing all three predicted HRE elements (Fig 1D). Within this construct, we created a series of ten consecutive small internal deletions within the CpG Island that have previously been identified as containing transcriptional regulatory elements (Foxler *et al*, 2011; Fig EV1I). These reporter constructs displayed ~ threefold induction of wild-type *LIMD1* promoter activity in hypoxia compared to normoxia. However, deletion of the 31-bp Δ3 region that encompasses the predicted HRE3 ablated any hypoxia-induced increase in luciferase activity (Fig EV1J). Furthermore, internal deletion of the three predicted HREs confirmed HRE3 to be the active hypoxia-responsive element within the *LIMD1* promoter (Fig 1E). The position and sequence of this HRE is also highly conserved, further supporting its functional importance (Fig 1F and G).

We next determined which HIF-α paralogue was involved in *LIMD1* regulation by combining the *LIMD1* promoter-driven luciferase reporters (Foxler *et al*, 2011) with shRNA-mediated knockdown of HIF-1α and HIF-2α. Depletion of HIF-1α, but not HIF-2α, prevented induction of *LIMD1* expression in hypoxia (Fig EV2A). This finding was corroborated by ChIP and EMSAs, which further demonstrated HIF-1 binding to the *LIMD1* promoter (Fig 2A and B). siRNA-mediated depletion of HIF-1α reduced LIMD1 protein and mRNA expression under hypoxic and, to a lesser extent, normoxic conditions in all cell lines examined (Figs 2C and D, and EV2B–E). LIMD1 depletion did not affect *HIF1A* or *HIF2A* mRNA expression, with the exception of an observed increase in *HIF2A* mRNA in HeLa cells under hypoxic conditions (Fig EV2F–I). The decrease in *LIMD1* expression in normoxia following si-HIF-1α demonstrates that HIF-1 activity is required for *LIMD1* expression in normoxia, an observation that has been previously described for other genes (Pillai *et al*, 2011). Furthermore, under hypoxic conditions HIF preferentially binds to gene loci that are already transcriptionally active to further activate their expression (Xia & Kung, 2009). Thus, these data show that under hypoxic conditions, HIF-1 binds the *LIMD1* promoter to increase its expression.

Under normoxic conditions, LIMD1 scaffolds PHD2 and VHL to enable efficient degradation of HIF-1α (Foxler *et al*, 2012). Given that *LIMD1* is a hypoxia-responsive gene, we next performed co-immunoprecipitation assays to assess whether the

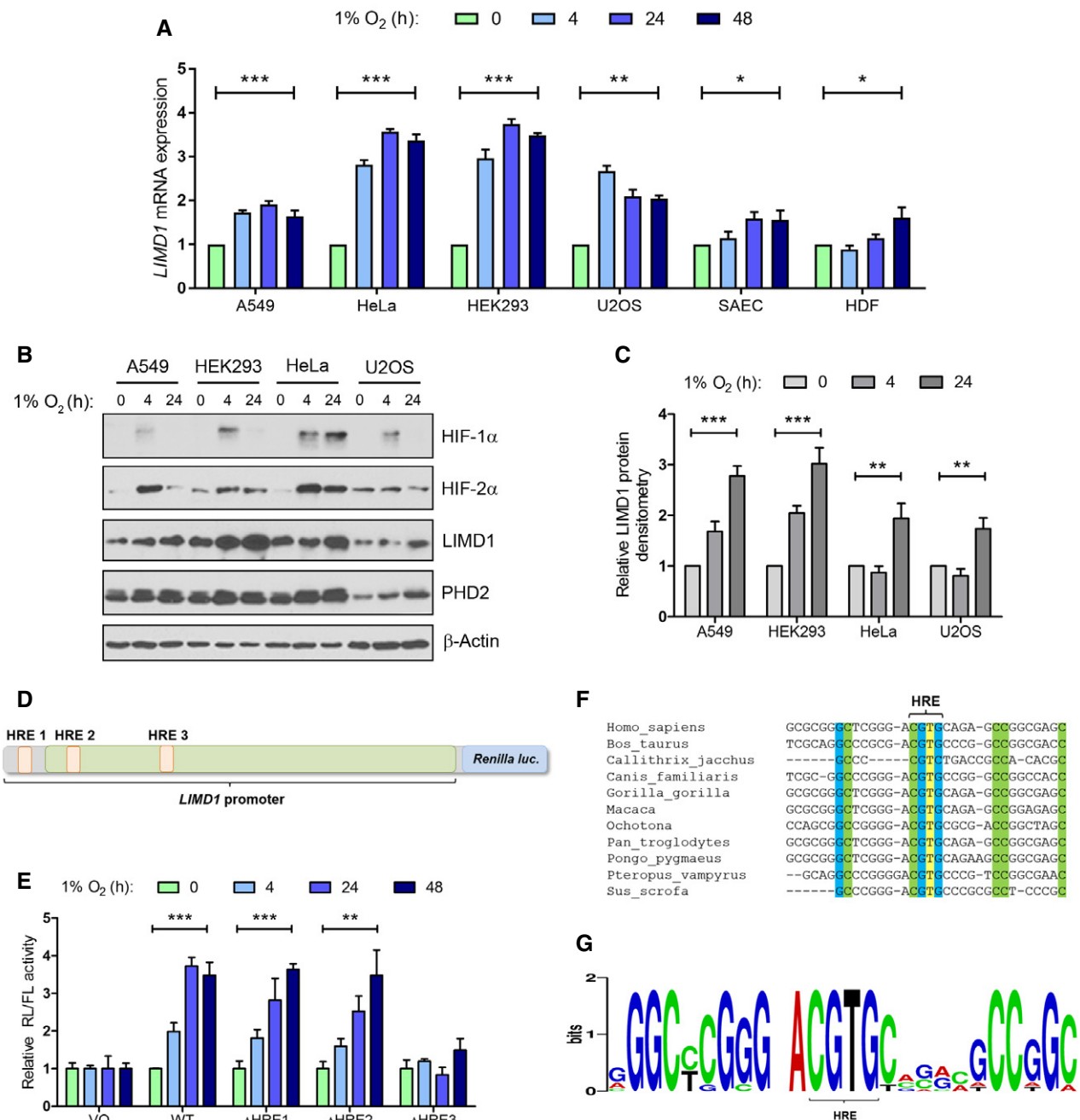

**Figure 1. *LIMD1* expression is regulated by hypoxia.**

The indicated panel of cell lines was exposed to either normoxia (20% $O_2$) or hypoxia (1% $O_2$) for up to 48 h prior to RNA and protein extraction.

A, B   (A) *LIMD1* mRNA and (B) protein levels were increased following hypoxic exposure.

C       Densitometric analysis of (B).

D       The *LIMD1* promoter contains a hypoxic response element responsible for HIF binding and transcriptional activation of *LIMD1*. Three predicted HRE elements were individually deleted within the context of the wild-type *LIMD1* promoter-driven Renilla luciferase.

E       Reporter constructs in (D) were expressed in U2OS cells and exposed to hypoxia for the indicated time-points. Luciferase activity was then assayed and normalised to firefly control. Data are displayed normalised to the normoxic value for each construct. Deletion of the third HRE present within the *LIMD1* promoter (ΔHRE3) inhibited hypoxic induction of *LIMD1* transcription.

F, G    (F) Sequence alignment and (G) sequence logo of *LIMD1* promoters from the indicated species demonstrate that the HRE3 consensus sequence is highly conserved.

Data information: Unless otherwise stated, data shown are mean ± SEM, $n = 3$, *$P < 0.05$, **$P < 0.01$, ***$P < 0.001$, according to the Student's *t*-test (A) or Holm–Šidák *post hoc* tests, comparing time-points within each cell line (A and C) or comparing the VO group to every other genotype within each time-point (E), following significant main effects/interactions of a mixed-model ANOVA. See Appendix Table S1 for a summary of statistical analysis.
Source data are available online for this figure.

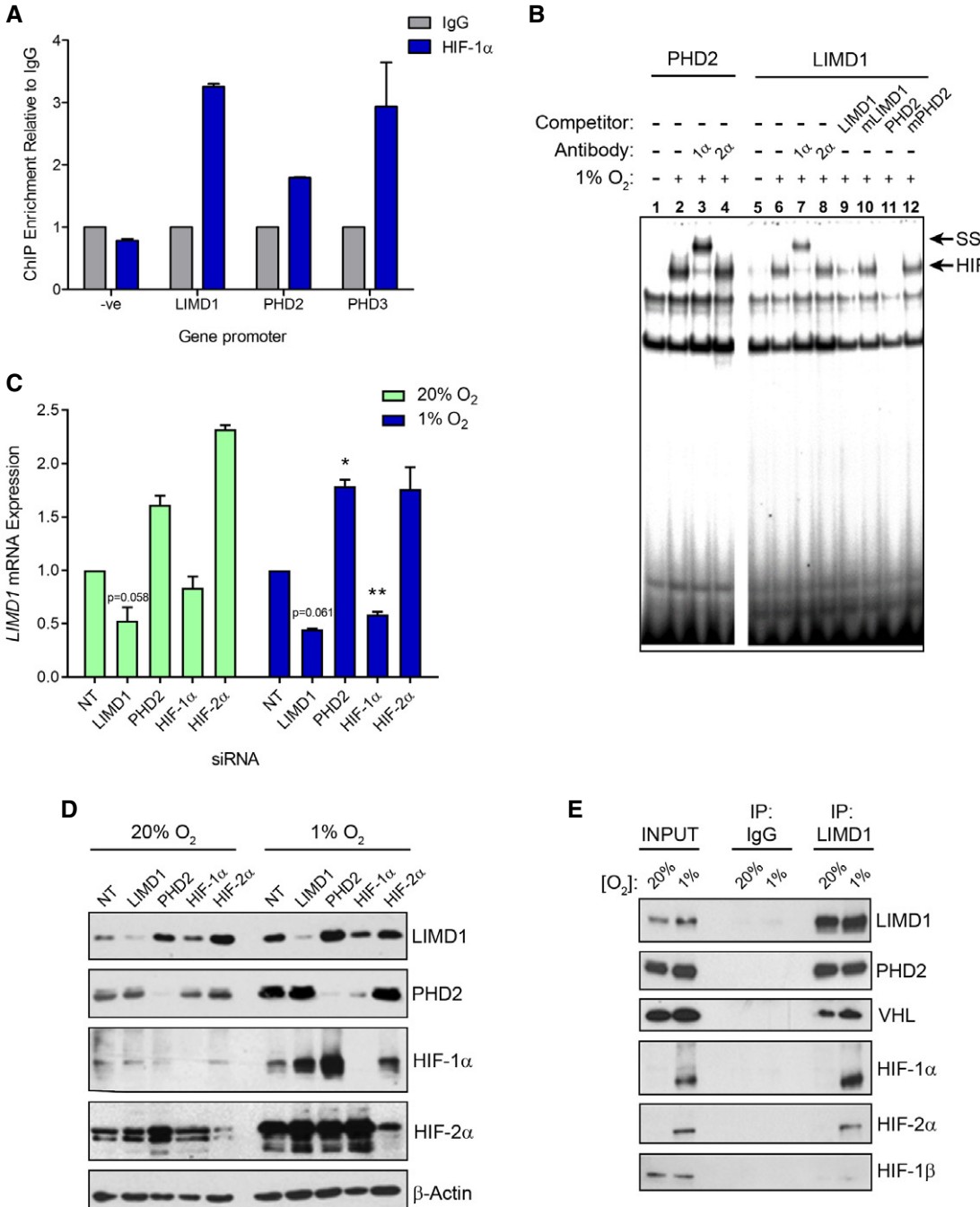

**Figure 2.  *LIMD1* is a HIF-1-responsive gene.**

A   HIF-1 binds to the *LIMD1* promoter. Chromatin immunoprecipitation assay (ChIP) of endogenous HIF-1α from paraformaldehyde cross-linked U2OS cells exposed to 16-h hypoxia, followed by qRT–PCR analysis of the indicated gene promoters.

B   EMSA of nuclear extracts from U2OS cells exposed to normoxia (lanes 1 and 5) or 16-h hypoxia identified that HIF-1α but not HIF-2α bound the *LIMD1* HRE consensus sequence, causing a band supershift (ss). Wild-type unlabelled oligo probes that encompass the *LIMD1* or *PHD2* HRE were used as controls to compete out the ss, and probes where the HRE sequences have been mutated (mLIMD1/mPHD2) were used to show specificity for HRE binding.

C   siRNA-mediated depletion of HIF-1α but not HIF-2α reduces LIMD1 expression in both normoxia and hypoxia. qRT–PCR analysis of *LIMD1* mRNA from U2OS cells transfected with the indicated siRNA (40 nM) and maintained in normoxia (20% O$_2$) or exposed to hypoxia (1% O$_2$) for 24 h.

D   Western blot analysis of protein from (C).

E   LIMD1 endogenously complexes with PHD2, VHL, HIF-1α and HIF-2α. Endogenous LIMD1 was immunoprecipitated from HeLa cells in either normoxia or following 24-h hypoxia and co-immunoprecipitated proteins identified by immunoblot analysis.

Data information: Unless otherwise stated, data shown are mean ± SEM, *n* = 3, *P < 0.05, **P < 0.01, according to Holm–Šidák-corrected one-sample Student's *t*-tests, comparing the mean of each gene's mRNA expression to the theoretical value of 1. See Appendix Table S2 for a summary of statistical analysis.
Source data are available online for this figure.

PHD2-LIMD1-VHL complex also exists in hypoxic conditions. Indeed, LIMD1 co-precipitated with PHD2 and VHL under hypoxia (Figs 2E and EV2J); HIF-1α and HIF-2α also co-precipitated with LIMD1, which may be due to the increased protein stability of the HIF proteins under this low oxygen tension. These data demonstrate active engagement of the PHD-LIMD1-VHL complex with its HIF target protein in hypoxia. However, HIF-1β did not co-precipitate within this complex, indicating LIMD1 was facilitating HIF-α degradation prior to heterodimerisation with the HIF-β subunit and independent of oxygen tensions. Thus, in hypoxia, LIMD1 expression facilitates formation of an active PHD2-LIMD1-VHL HIF-degradation complex.

## HIF-1-driven LIMD1 expression is required for negative regulation of HIF in a hypoxic environment

LIMD1 protein expression has been previously shown to be significantly reduced or lost in human lung and breast cancers (Sharp *et al*, 2004, 2008; Spendlove *et al*, 2008). This led us to hypothesise that a decrease in the normal levels of LIMD1 protein expression as a result of *LIMD1* loss of heterozygosity (LOH) or promoter methylation (Sharp *et al*, 2008) may disrupt the hypoxic PHD-LIMD1-VHL complex, and exacerbate HIF-mediated gene expression and pro-transforming effects in the context of a hypoxic tumour microenvironment.

To directly assay the effects of hypoxia-driven LIMD1 expression, we utilised a lentiviral shRNA-mediated knockdown-rescue system that concurrently expresses an shRNA and a cDNA (Foxler *et al*, 2012; Fig 3A schematic). We utilised this system to express an sh*LIMD1* to deplete cells of endogenous LIMD1, whilst simultaneously re-expressing an RNAi-resistant (rr) Flag epitope-tagged LIMD1 that was under the control of the endogenous *LIMD1* promoter, which we previously identified as being an active and regulated promoter sequence under both normoxic and hypoxic conditions (Fig EV1I; Foxler *et al*, 2011). The promoter sequence contained the wild-type HRE motif (HRE[wt]); we also generated a version with a mutated HRE sequence (HRE[mut]; Figs 3A and EV3A). Ectopic *LIMD1* expression in cells transduced with these vectors (where endogenous *LIMD1* is repressed by the shRNA) would therefore be potentially enhanced (HRE[wt]) or unchanged (HRE[mut]) by hypoxia through HIF-1.

U2OS, HeLa and SAEC were transduced with the paired set of lentiviruses described and identified within this non-clonal population, and *LIMD1* controlled by the HRE[wt] promoter had a twofold to threefold hypoxic induction of LIMD1 (Figs 3B–E and EV3B–E). In contrast, mutation of the HRE within the *LIMD1* promoter (HRE[mut]) significantly impaired hypoxic induction of LIMD1 in these lines. This was coupled with an impairment of HIF-1α degradation under increasing exposure to hypoxia in the HRE[mut] lines compared to HRE[wt] (Fig 3B and F–H). Of note, the HRE[mut] cells had increased expression of *HIF1A* mRNA after 24 h in hypoxia compared to the HRE[wt] cells (Fig EV3F), which we postulate may be the result of increased HIF-1α protein in this line further driving its own transcription (Koslowski *et al*, 2011). The HRE[mut] cell lines also exhibited increased HIF-driven luciferase activity (Figs 3I and EV3G), endogenous HIF-1-driven gene activation (Figs 3J and EV3H–K) and cumulative secreted VEGF-A (Figs 3K and EV3L) when compared to the HRE[wt] cells.

We then wished to ascertain whether the observed transcriptional and phenotypic differences (Fig 3) were both HIF-specific and due to the effects that LIMD1 expression was exerting on HIF-1α protein turnover. Treatment with the translational inhibitor cycloheximide (Cx) to assess the degradation rate of HIF revealed the HRE[mut] line had a significantly decreased rate of both HIF-1α and HIF-2α degradation compared to the HRE[wt] line (Fig 4A and B). Furthermore, siRNA depletion of HIF-1α ablated the differential gene expression of VEGF-A between the lines (Fig 4C and D). Depletion of HIF-2α also decreased VEGF-A expression, likely due to *VEGF-A* being a dual HIF-1 and HIF-2 target gene (Hu *et al*, 2003). We conclude that the differences in HIF stability and transcription observed between the HRE[wt] and HRE[mut] cell lines were caused by the hypoxic induction of LIMD1 via its HRE.

To exclude the possibility that the miRNA-silencing function of LIMD1 (James *et al*, 2010) was complicit in this observed effect, we utilised luciferase reporter constructs containing the HIF-1/2α 3′UTRs (as defined from RefSeq identifiers NM_001530.3 and NM_001430.4, respectively). There were no differences in HIF-α 3′ UTR regulation in LIMD1-expressing or null cells, regardless of oxygen tension (Fig 4E), indicating that LIMD1 was only affecting HIF-α levels post-translationally and not post-transcriptionally in this experimental context of subtle but significant change in LIMD1 protein induction [twofold to threefold increase (Fig 3B–E)].

---

**Figure 3.  Induction of LIMD1 in hypoxia inhibits HIF-1-mediated gene expression.**

A     A combinatorial RNAi–cDNA re-expression lentiviral cassette was utilised to create isogenic cell lines where *LIMD1* was either responsive or unresponsive to hypoxia. Endogenous LIMD1 was depleted by shRNA, whilst concurrently a Flag-LIMD1 cDNA was expressed that was driven by the sequence of its own endogenous promoter.

B     U2OS, HeLa and SAEC were transduced with these lentiviral cassettes to create the HRE[wt] and HRE[mut] paired cell lines.

C–E   Western blot quantification of LIMD1 relative to β-actin and normalised to 0-h time-point for each cell line.

F–H   Western blot quantification of HIF-1α relative to β-actin and normalised to 4-h time-point for each cell line.

I     Impaired hypoxic induction of *LIMD1* induction increases HRE-luciferase activity. U2OS isogenic cell lines were co-transfected with a synthetic HRE-luciferase (pNL-HRE) and pGL3 firefly normalisation plasmid, prior to exposure to hypoxia. Luciferase activity was assayed and normalised against HRE activity in the HRE[wt] line. After 24-h hypoxic exposure, the HRE[mut] line had significantly increased luciferase activity compared to the HRE[wt] line.

J     Impaired hypoxic induction of *LIMD1* induction increases expression of HIF target genes. RNA was extracted from the U2OS isogenic cell lines following 24-h hypoxic exposure, and a panel of HIF-1 downstream targets were quantified by qRT–PCR. The HRE[mut] line had significantly increased HIF-1-driven gene expressions compared to the HRE[wt] line.

K     Impaired hypoxic induction of *LIMD1* induction increases VEGF-A secretion. U2OS isogenic cell lines were incubated in hypoxia for 48 h and VEGF-A secretion was quantified by ELISA, identifying the HRE[mut] line as secreting a significantly increased VEGF-A protein when compared to the HRE[wt] line.

Data information: Unless otherwise stated, data shown are mean ± SEM, *n* = 3, *P < 0.05, **P < 0.01, ***P < 0.001, according to Holm–Šidák *post hoc* tests, comparing genotypes at each time-point (I and K) or at each gene (J), following significant main effects/interactions of a mixed-model ANOVA. See Appendix Table S3 for a summary of statistical analysis.

Source data are available online for this figure.

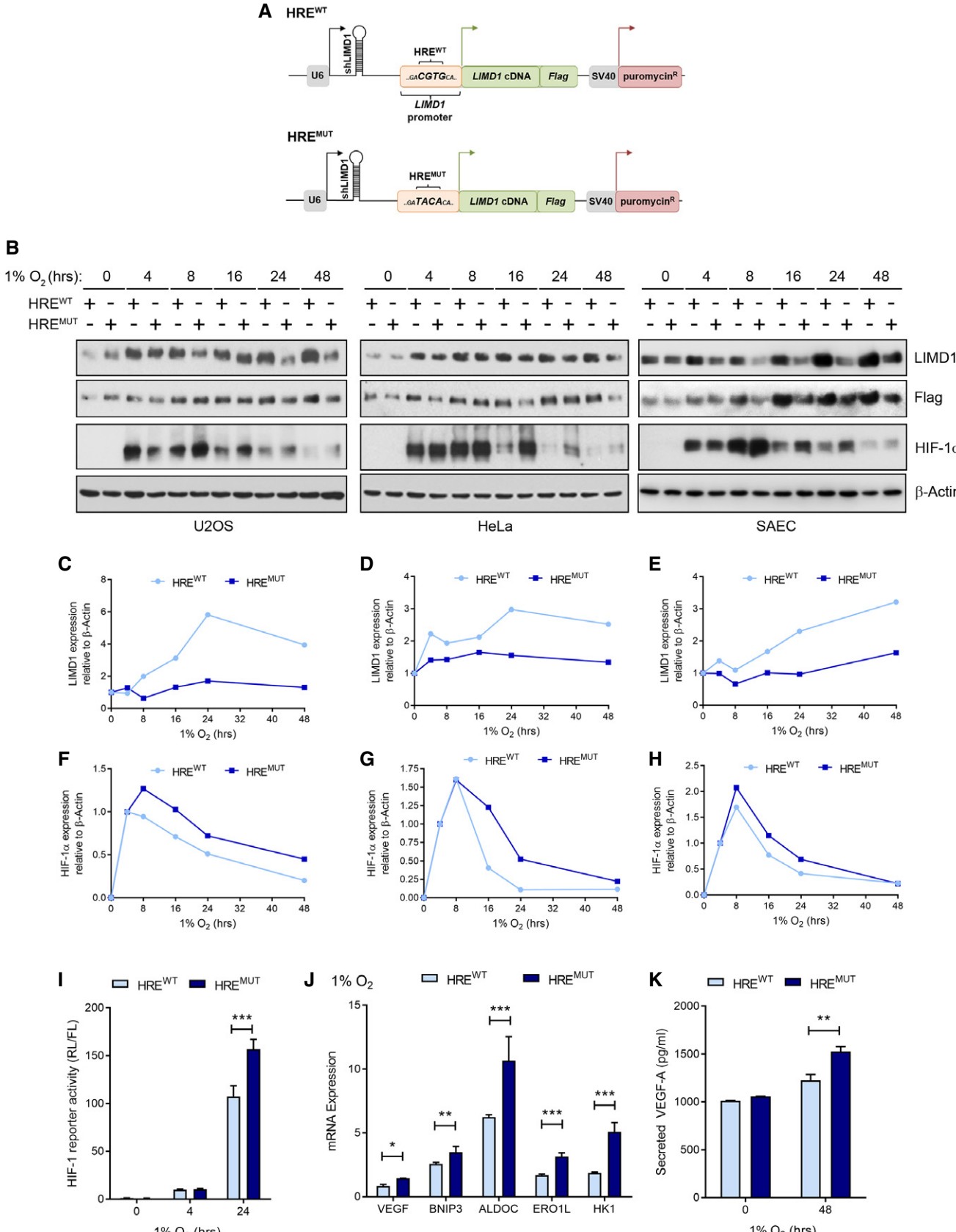

Figure 3.

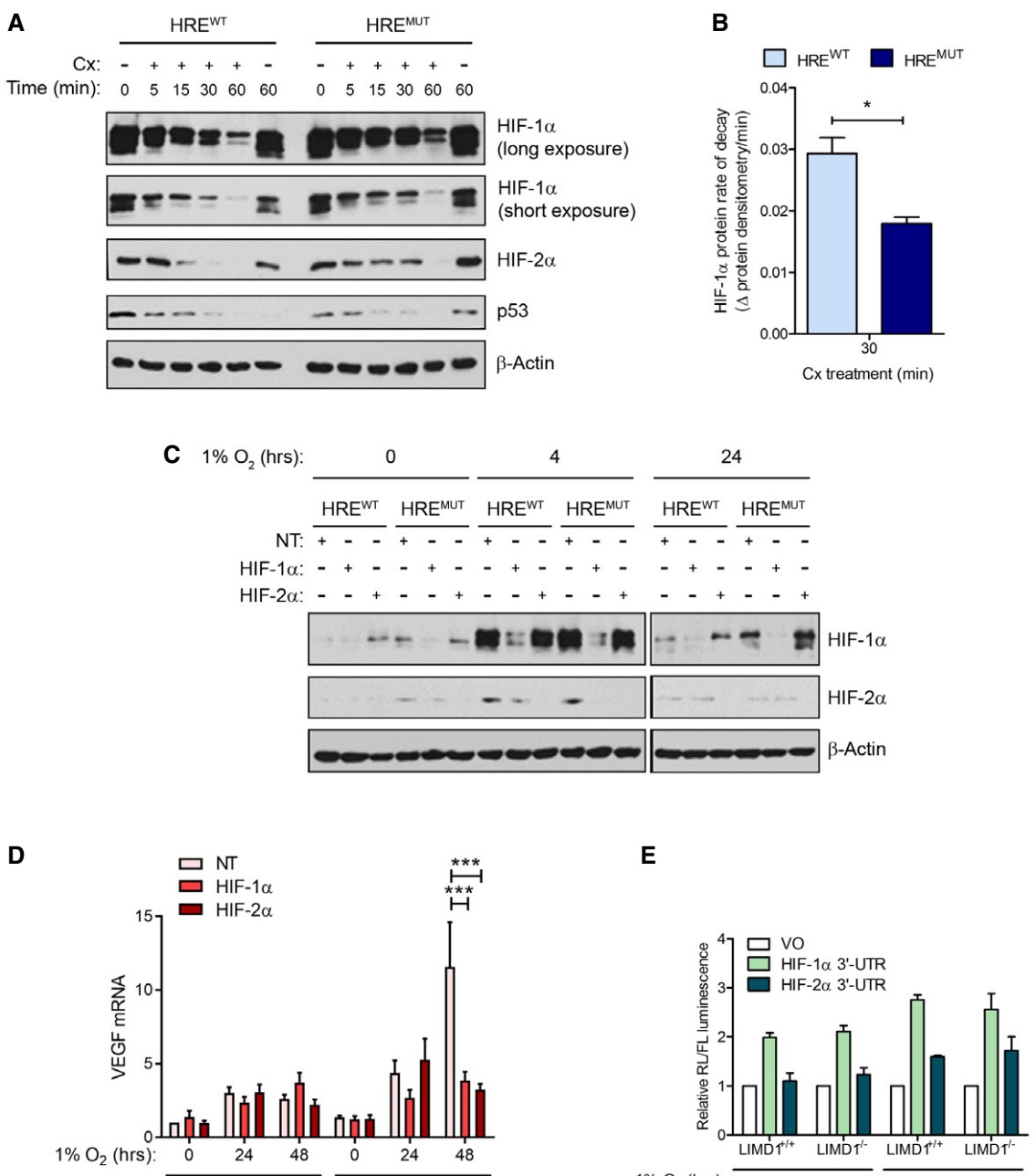

**Figure 4. LIMD1 regulates HIF-1α at the post-translational level.**

A    The hypoxic induction of LIMD1 facilitates HIF protein degradation. Western blot analysis of HRE[wt] and HRE[mut] U2OS lines exposed to hypoxia and the translation inhibitor cycloheximide (Cx) treatment for the indicated time-points. HIF-1α protein is degraded more efficiently in the HRE[wt] line where *LIMD1* expression is increased in hypoxia as detected by immunoblot.

B    HIF-1α rate of decay (ROD) is significantly slower in HRE[mut] U2OS compared to HRE[wt]. Densitometric analysis of HIF-1α protein, double normalised to β-actin and 0 min Cx treatment in each line, was used to calculate ROD (Δ relative protein densitometry per minute) of HIF-1α protein between 0 and 30 min of Cx treatment.

C, D   (C) The increase in HIF target gene expression in the HRE[mut] lines is due to HIF protein expression. siRNA (40 nM) targeting HIF-1α and HIF-2α was transfected into the HRE[wt] and HRE[mut] cell lines, and protein depletion was confirmed by immunoblot, which resulted in decreased expression of *VEGF* mRNA as analysed by qRT–PCR from the simultaneous extraction of RNA (D).

E    LIMD1 expression does not affect miRNA silencing of *HIF-1α* and *HIF-2α* 3′UTR in normoxia or hypoxia. The 3′UTR of either HIF-1α or HIF-2α was cloned into the psiCheck2 luciferase vector. These were then transfected into LIMD1-expressing control (LIMD1[+/+]) or LIMD1-knockout (LIMD1[−/−]) HeLa cells and luciferase activity assayed. Expression of LIMD1 did not affect the stability or silencing of either the HIF-1α or HIF-2α 3′UTR-containing reporters.

Data information: Unless otherwise stated, data shown are mean ± SEM, n = 3, *P < 0.05, ***P < 0.001, according to the Student's *t*-test (B) or Holm–Šidák *post hoc* tests, comparing siRNA treatments within each combination of genotype and time, following significant main effects/interactions of a mixed-model ANOVA. See Appendix Table S4 for a summary of statistical analysis.

Source data are available online for this figure.

    

Together, these data reveal that inhibition of the HIF-1/LIMD1 feedback loop causes an increased cellular hypoxic HIF phenotype *in vitro*, demonstrating that increased LIMD1 protein expression in hypoxia is necessary for correct modulation of HIF-1 expression and signalling in a hypoxic environment.

### Ablation of HIF-driven LIMD1 expression promotes tumour growth

We next investigated the physiological relevance of this newly identified hypoxic HIF-1–LIMD1 negative feedback loop using *in vivo* xenograft tumour growth as a model system. LIMD1 is a lung tumour suppressor, with decreased mRNA and protein expression shown to occur in a large proportion of lung adenocarcinomas (Sharp *et al*, 2008). For our xenograft model, we therefore utilised the A549 lung adenocarcinoma cell line. A549 LIMD1-HRE$^{wt}$ and A549 LIMD1-HRE$^{mut}$ cell lines were created as described above. The transduced cell lines were validated *in vitro*, where ablation of the hypoxic induction in *LIMD1* expression in the HRE$^{mut}$ line increased synthetic HIF-1-driven luciferase activity, HIF-1-responsive genes and secreted VEGF (Fig 5A–C). These findings also corroborated the results obtained in U2OS and HeLa cells (Figs 3 and EV3).

Subcutaneous xenografts were established on the flanks of SCID/beige mice from either the A549 LIMD1 HRE$^{wt}$ or HRE$^{mut}$ cell lines. Xenografts from A549 HRE$^{mut}$ cells (which have an impaired HIF–LIMD1 negative feedback loop) had increased age-matched endpoint tumour volumes compared to A459 HRE$^{wt}$ cells (which have an intact HIF–LIMD1 negative feedback loop; Fig 5D). The effect on *in vivo* tumour growth was not due to intrinsic differences in proliferation rates as HRE$^{wt}$ and HRE$^{mut}$ cells had similar growth rates in *in vitro* proliferation assays and in colony formation assays under either normoxia or hypoxia (Fig EV4A and B). Endomucin staining, as a marker of blood vessels, revealed increased vasculature in HRE$^{mut}$ A549-derived xenografts (Fig 5E and F) and was associated with increased *endomucin* mRNA expression (Fig 5G) along with an increased HIF-1-mediated gene expression profile of pro-angiogenic and glycolytic genes (Figs 5H–

J and EV4C–O). *HIF1A* mRNA expression was not altered upon LIMD1 loss (Fig EV3F), demonstrating that the increase in HIF-1-driven gene expression was not due to increased *HIF-1α* transcription. Together, these data indicate that ablation of this HIF-1–LIMD1 negative regulatory feedback mechanism *in vivo* increases tumour growth and vascularisation.

### LIMD1 is a prognostic indicator in NSCLC

Finally, to investigate the clinical relevance and significance of our *in vitro* and *in vivo* findings in NSCLC, we examined LIMD1 protein expression in a tissue microarray of 276 lung cancer patients and investigated correlation with patient outcome (representative staining Fig 6A; marker distribution Fig 6B). In agreement with previous studies, LIMD1 protein expression was detected in both nuclear and cytoplasmic compartments (Sharp *et al*, 2008; Spendlove *et al*, 2008). Kaplan–Meier survival analysis demonstrated that patients exhibiting low LIMD1 expression within this cohort had significantly worse overall survival compared to those with high LIMD1 expression ($P = 0.045$; Fig 6C). Immunohistochemical analysis of HIF-1α and downstream angiogenic marker VEGF-A was not significantly correlated with LIMD1 expression in this cohort; however, high VEGF expression was correlated with poor patient prognosis ($P = 0.045$; Appendix Fig S1, Fig EV5A–C).

We next interrogated The Cancer Genome Atlas (TCGA) datasets to assess *LIMD1* loss in a much larger cohort of NSCLC. First, gene copy number analysis of *LIMD1* and a number of other well-characterised tumour suppressor genes in lung adenocarcinoma (LUAD) and squamous cell carcinoma (LUSC) cohorts ($n = 512$ and $n = 498$, respectively) demonstrated that single (shallow) or bi-allele (Ghosh *et al*, 2008) deletion of the *LIMD1* gene occurred in 47.1% (LUAD) and 85.4% of patients (LUSC; Figs 6D and EV5D). Regression analysis demonstrated correlation between *LIMD1* copy number and reduced mRNA expression (Fig EV5E and F); therefore, lung adenocarcinoma patients were stratified into risk groups (quartiles) based on mRNA abundance intensities, and patient survival was determined using a Cox proportional hazards model. We

---

**Figure 5. Increased LIMD1 expression in hypoxia inhibits tumour growth and vascularisation.**

A   Impaired hypoxic induction of *LIMD1* induction increases HRE-luciferase activity in lung adenocarcinoma cells. Isogenic HRE$^{wt}$ and HRE$^{mut}$ A549 cell lines were co-transfected with a synthetic HIF-1-driven luciferase (pNL-HRE) and pGL3 firefly normalisation plasmid, prior to exposure to hypoxia. Luciferase activity was assayed and normalised against HRE activity in the HRE$^{wt}$ line.

B   Impaired hypoxic induction of *LIMD1* induction increases endogenous HIF-mediated gene expression. RNA was extracted from the isogenic cell lines following 24-h hypoxic exposure, and the expression of a panel of HIF-1 target genes was quantified by qRT–PCR. The HRE$^{mut}$ line had significantly increased HIF-1-driven gene expression compared to the HRE$^{wt}$ line.

C   Impaired hypoxic induction of *LIMD1* induction increases VEGF-A secretion in lung adenocarcinoma cells. The isogenic cell lines were incubated in hypoxia for 48 h, and VEGF secretion was quantified by ELISA, identifying the HRE$^{mut}$ line as secreting a significantly increased level of VEGF-A protein when compared to the HRE$^{wt}$ line.

D   Impaired hypoxic induction of *LIMD1* induction increases 3D tumour growth *in vivo*. Eight- to 12-week-old female SCID/beige mice were injected subcutaneously with $5 \times 10^9$ A549 HRE$^{wt}$ or HRE$^{mut}$ cells and subsequent xenograft growth measured. HRE$^{wt}$-derived xenografts were smaller in volume compared to the HRE$^{mut}$ where LIMD1 expression was unresponsive to hypoxia.

E   HRE$^{wt}$-derived xenografts have lower blood vessel density compared to HRE$^{mut}$-derived xenografts. Xenografts were snap frozen in liquid nitrogen, sectioned and stained with endomucin (red) and DAPI (blue) (upper panels). Lower panels show endomucin staining in white for visual clarity. Scale bar, 100 μm.

F   Blood vessels were manually counted throughout the entire section and xenograft cross-sectional area calculated.

G–J   (G) HRE$^{wt}$-derived xenografts have lower expression of the blood vessel marker *endomucin*. RNA was extracted from snap-frozen xenografts and analysed by qRT–PCR. In addition, HRE$^{wt}$ xenografts also had lower expression of the HIF-driven genes (H) *VEGF-A*, (I) *HK1* and (J) *PDK1*. $n = 15$ (HRE$^{wt}$) and $n = 14$ (HRE$^{mut}$).

Data information: Unless otherwise stated, data shown are mean ± SEM, $n = 3$, *$P < 0.05$, **$P < 0.01$, ***$P < 0.001$, according to Holm–Šidák *post hoc* tests, comparing genotypes at each time-point (A and C) or at each gene (B), following significant main effects/interactions of a mixed-model ANOVA (A and C) or standard two-way ANOVA (B); alternatively, data in panel (D) were analysed with a Mann–Whitney *U*-test, and panels (F-J) were analysed with separate Welch-corrected Student's *t*-tests. See Appendix Table S5 for a summary of statistical analysis.
Source data are available online for this figure.

determined that patients in the risk group exhibiting high *LIMD1* expression (High Exp) had increased overall survival, whereas patients exhibiting low *LIMD1* expression (Low Exp) had reduced overall survival (log-rank $P = 0.021$, HR 0.6; Fig 6E).

To assess the impact of *LIMD1* loss on HIF regulation and outcome in these patients, we analysed the TCGA LUAD cohort to identify hypoxia/HIF signature genes correlated with low LIMD1 expression. We identified a strong inverse correlation between *LIMD1* and HIF target genes *SLC2A1*, *GAPDH* and *IGF2BP2* mRNA expression (Fig 6F–H). Kaplan–Meier analysis of survival of patients stratified by expression of these genes revealed that patients with the highest expression of these genes have significantly worse overall survival compared to patients who demonstrate the lowest expression (Fig 6I–K).

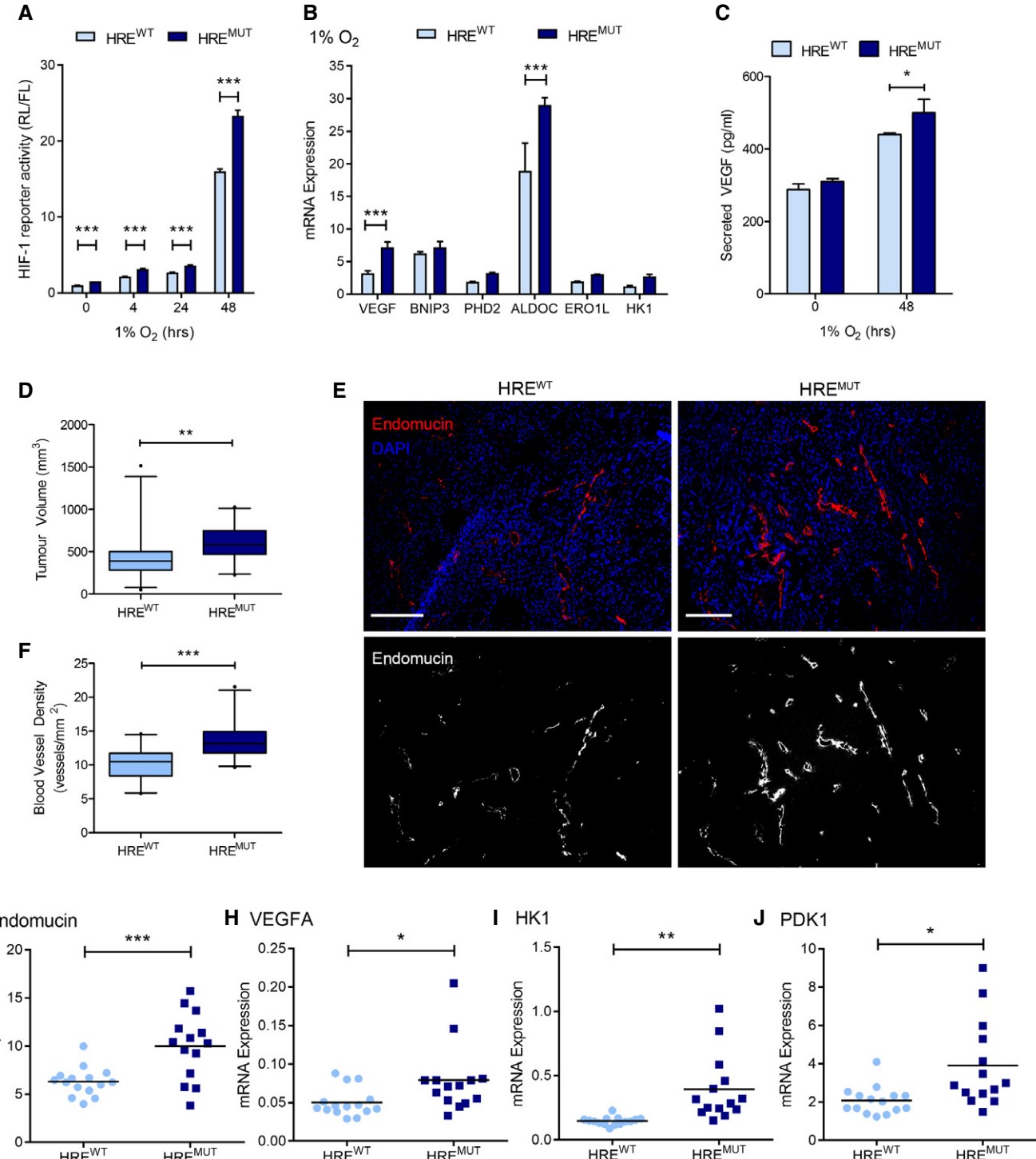

**Figure 5.**

To complement the bioinformatic analysis of patient tumours, we performed cell line-based *in vitro* analysis of the gene expression changes that occur in a primary lung epithelial cell background following a reduction in LIMD1 expression. We used siRNA to knock down *LIMD1* in primary human lung bronchial epithelial cells (HBEC) and performed micro-array analysis to identify the resultant gene ontology changes. Due to the cell type used in this analysis, this models the role of LIMD1 loss in lung SCC. Differentially expressed genes and associated pathways following depletion of LIMD1 were analysed by ingenuity pathway analysis (IPA), and this identified upregulation of HIF-1α and a network of HIF-1 interactions following LIMD1 loss (Appendix Figs S2 and S3, and Dataset EV1).

In summary, these findings reveal the existence of a previously uncharacterised tumour-suppressive, negative regulatory feedback loop in which LIMD1 facilitates HIF degradation in hypoxia (Fig 7). These findings add another level to the rapidly growing number of pathways and processes that regulate HIF. This is the first example of the main scaffold protein LIMD1 within the regulatory PHD2-LIMD1-VHL complex being itself regulated by HIF and therefore providing this regulatory triad with an intrinsic homeostatic negative regulatory functionality, which when deregulated contributes to lung adenocarcinoma tumour growth.

# Discussion

We have previously shown that the scaffold protein LIMD1 is a critical component of the HIF-degradation complex (Foxler *et al*, 2012). In this study, we have demonstrated that LIMD1 expression is induced by HIF-1 under hypoxic conditions, forming a negative feedback loop to degrade HIF. HIF-1-driven LIMD1 expression enables the cell to degrade HIF-1α under conditions of chronic hypoxia to prevent protracted HIF-1 activation, frequently associated with an oncogenic phenotype. In a xenograft model for tumorigenesis, ablation of the hypoxic inducibility of LIMD1 expression and subsequent loss of hypoxic HIF-1α protein regulation caused increased tumour vasculature and growth. From a clinical perspective, decreased LIMD1 expression correlates with increased expression of HIF target genes *SLC2A1*, *GAPDH* and *IGF2BP2*, each of which correlates with significantly worse survival outcomes for patients.

By virtue of its scaffold function, the cellular processes and pathways that LIMD1 regulates are dependent on its protein partners. HIF-α is post-transcriptionally regulated (Gorospe *et al*, 2011), including by microRNA-20b, microRNA-199a and microRNA-424 (Rane *et al*, 2009; Cascio *et al*, 2010; Ghosh *et al*, 2010a). Therefore to rule out the possibility that the miRNA-silencing function of

LIMD1 (James *et al*, 2010) was contributing to the observations made, we identified that loss of LIMD1 does not affect the stability nor silencing of *HIF1A/HIF2A* mRNA. LIMD1's multiple tumour-suppressive functions and its discrimination between binding partners are likely to be regulated by different signalling cascades, with multiple phosphorylation events on LIMD1 having already been identified (Huggins & Andrulis, 2008; Sun & Irvine, 2013). However, any post-translational modification events that promote LIMD1 to function preferentially in HIF regulation are yet to be elucidated, but may stem from a hypoxia-/HIF-activated signalling cascade (Wouters & Koritzinsky, 2008; Lee *et al*, 2012; Xu *et al*, 2017).

LIMD1 has few reported coding sequence mutations and none that correlate with loss of function (Huggins *et al*, 2007; Ghosh *et al*, 2008). Such data suggest that gene dosage and thus small but significant changes in protein levels may be important in disease aetiology. Loss of total gene expression is frequently observed, where reduced *LIMD1* gene copy number and mRNA expression occur in a significant proportion of lung carcinomas (Sharp *et al*, 2008). In this study, we have identified that a reduction in *LIMD1* expression (through ablation of the hypoxic responsiveness of *LIMD1* promoter) is sufficient to cause a HIF-mediated pathological transcriptome and phenotype in the form of increased tumour size and vasculature.

The regulation of HIF under normoxic conditions is well characterised (Salceda & Caro, 1997; Maxwell *et al*, 1999); however, the mechanism of HIF degradation under long-term hypoxic conditions is still poorly defined at the molecular level, even less so with respect to the physiological relevance. However, our new data show that discrete regulatory processes such as the HIF/LIMD1 negative feedback loop described here can modulate HIF activity under chronic hypoxic conditions. Biological pathways often have in-built negative feedback loops whereby a transcription factor induces the expression of an upstream negative regulator (Yosef & Regev, 2011); such a negative feedback loop exists for the HIF proteins and the hypoxic response (Marxsen *et al*, 2004; Stiehl *et al*, 2006; Tan *et al*, 2008; Nakayama *et al*, 2009). PHD2 depletion under hypoxia results in stabilisation of HIF-1α (Stiehl *et al*, 2006), demonstrating HIFs can be actively degraded and regulated independent of oxygen tensions. Indeed, the complexity of HIF regulation via the hydroxylation–ubiquitination degradation pathway is becoming increasingly clear, with the identification of a plethora of HIF regulators, more recently including Siah2, SHARP1 and RHOBTB3 (Nakayama *et al*, 2009; Montagner *et al*, 2012; Zhang *et al*, 2015).

Expression of HIF-α mRNA has historically been described as constitutive; however, a growing number of studies are demonstrating the existence of factors that regulate HIF-α mRNA (Uchida *et al*,

---

**Figure 6. Loss of LIMD1 expression is a poor prognostic indicator in lung cancer.**

A, B Representative immunohistochemistry (IHC) staining of cores for LIMD1 that were scored in the 276 patient sample TMA to ascertain relative expression (H-score) within the cohort (B). Scale bar 100 μm; 20 μm on zoom panel.

C Kaplan–Meier analysis identified that patients stratified as having low LIMD1 expression (weak staining) exhibit poorer overall survival compared to high (intense staining).

D Copy number alterations of *LIMD1* and other validated tumour suppressor genes from a lung adenocarcinoma (LUAD) cohort, publically available from TCGA.

E Stratification of TCGA LUAD cohort into quartiles based on *LIMD1* mRNA expression (where Q1 = lowest expression quartile, Q3 = highest expression quartile) demonstrates worse overall survival for patients within the lowest *LIMD1*-expressing quartile (Low Exp) compared to the highest *LIMD1*-expressing quartile (High Exp).

F–H (F) Correlation analysis of *LIMD1* mRNA expression in patients from (D) identified a significant inverse correlation between *LIMD1* and HIF target genes *SLC2A1*, (G) *GAPDH* and (H) *IGF2BP2*. The violin plots show the local density of data: black bars represent the IQR, red dot is the mean and white dot is the median.

I–K (I) Stratification of TCGA LUAD cohort into quartiles based on *SLC2A1*, (J) *GAPDH* and (K) *IGF2BP2* expression demonstrates worse overall survival for patients within the highest expressing quartile (High Exp) compared to the lowest expressing quartile (Low Exp) for each gene.

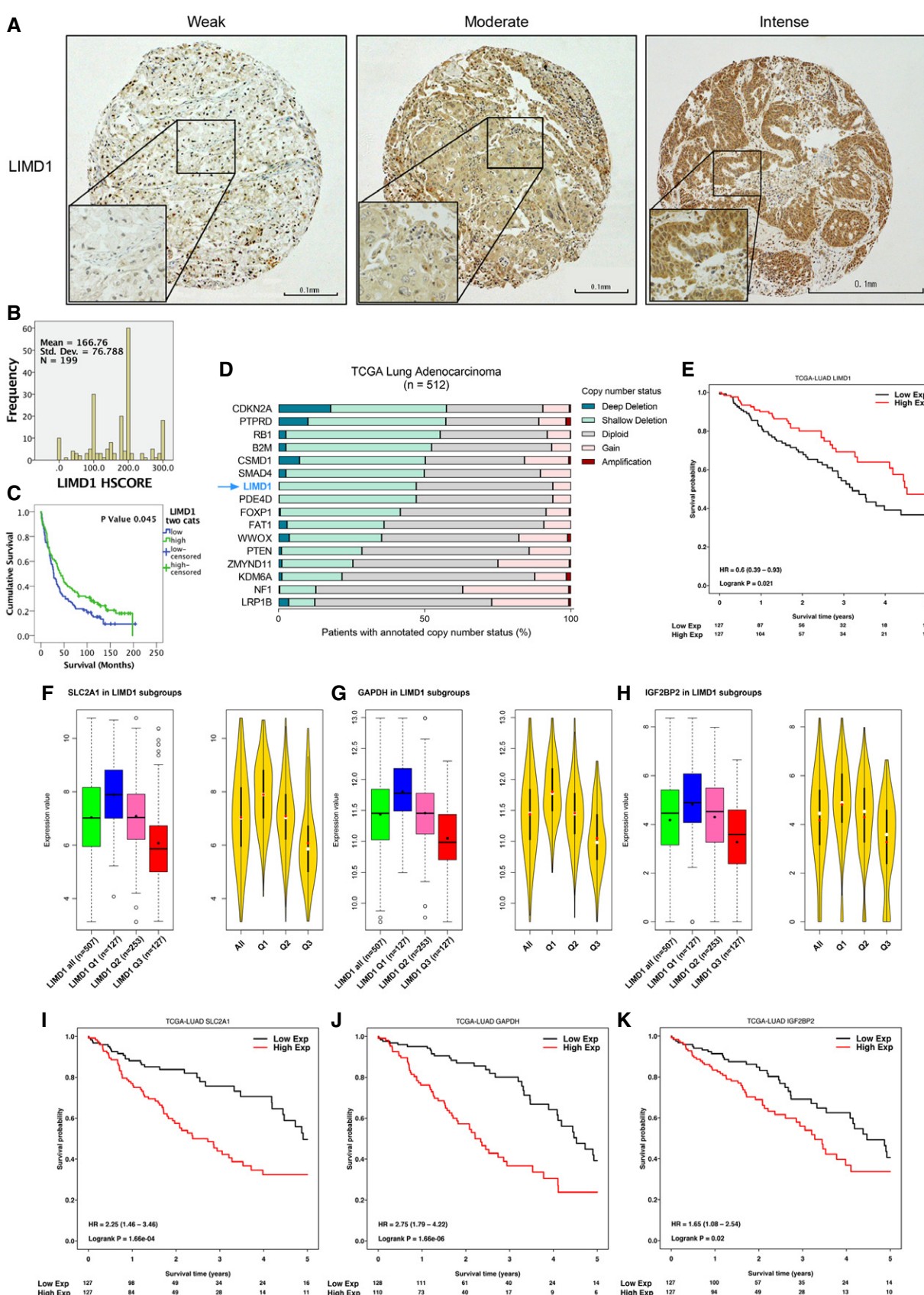

**Figure 6.**

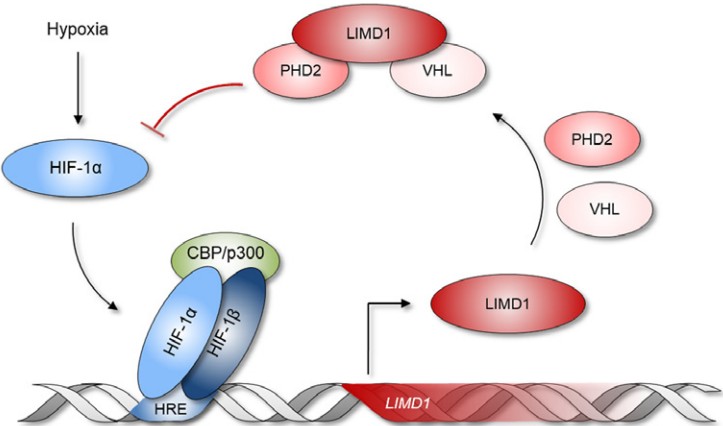

**Figure 7. Proposed model of the HIF-1–LIMD1 negative regulatory feedback mechanism.**

HIF-1α is stabilised in hypoxia and heterodimerises with HIF-1β, forming an active HIF-1 transcription factor complex with CBP/p300. HIF-1 binds to hypoxic response elements (HRE) within the promoters of genes that are required for hypoxic survival and adaptation, increasing their gene expression. Classically, these include genes required for metabolic adaptation and angiogenesis. Here, we identify that the tumour suppressor gene *LIMD1* is also a HIF-1-responsive gene, and in hypoxia, its expression is increased at both the mRNA and protein levels. This increased expression facilitates formation of a hypoxic PHD2-LIMD1-VHL degradation complex, facilitating hypoxic HIF-1α degradation. In turn, this attenuates the tumorigenic cellular adaption to hypoxia and subsequent tumorigenesis, thus identifying a new level of complexity of a tumour-suppressive mechanism of LIMD1.

2004; Chamboredon *et al*, 2011; Hamidian *et al*, 2015). From our investigations, the *LIMD1* HRE[mut]-derived A549 xenografts had an increase in *HIF2A* mRNA expression compared to the HRE[wt] controls. We propose that inhibition of hypoxia-driven *LIMD1* expression, resulting in enhanced HIF-α protein stability, drives *HIF2A* mRNA expression. Stabilisation of HIF-α protein and hypoxia has previously been described as inducers of *HIF2A* mRNA expression (Hamidian *et al*, 2015; Mohlin *et al*, 2015).

Many HIF-activated genes have been identified as prognostic and diagnostic markers. The oncogenic properties of HIF in cancer and disease have resulted in an abundance of potential therapeutics in development and clinical trials that target HIF at the transcriptional, translational, post-translational and functional levels (Masoud & Li, 2015; Nakazawa *et al*, 2016). Indeed, the hypoxic HIF signalling pathway is already a therapeutic target with multiple drugs currently in clinical trials (Wilson & Hay, 2011; Bryant *et al*, 2014; Masoud & Li, 2015). VEGF expression is a marker of poor prognosis in NSCLC (Kaya *et al*, 2004); as such, VEGF-targeted therapies are widely used to target VEGF-mediated angiogenesis, although the details of how they exert their effects are not yet clearly defined (Ellis & Hicklin, 2008). In some cases, vasculature promotion and normalisation have been demonstrated to yield greater therapeutic efficacy (Wong *et al*, 2015, 2016). In our small cohort of 276 NSCLC, high VEGF expression was correlated with poor survival, with 80% of the cohort demonstrating high VEGF expression.

Analysis of a TCGA lung adenocarcinoma cohort determined an inverse correlation between expression of LIMD1 and HIF target genes *SLC2A1*, *GAPDH* and *IGF2BP2*. Overexpression of glucose transporter 1 (GLUT-1), the uniporter protein encoded by the *SLC2A1* gene, is correlated with poor survival in most solid tumours (Wang *et al*, 2017), and selective GLUT-1 inhibition has been demonstrated to inhibit glucose uptake and reduce viability of lung adenocarcinoma and squamous cell carcinoma cell lines *in vitro* and *in vivo* (Goodwin *et al*, 2017). Correlative studies have also determined upregulation or *de novo* expression of the IGF2BP family of oncofetal proteins across a number of solid tumours to be associated with tumour aggressiveness, metastasis and poorer overall survival (Bell *et al*, 2013). GAPDH overexpression is also associated with reduced patient survival (Guo *et al*, 2013), and therapeutic targeting of GAPDH has been demonstrated to have clinical application in both hepatocellular and colorectal cancers (Ganapathy-Kanniappan *et al*, 2012; Yun *et al*, 2015). Our data correlate *LIMD1* loss with overexpression of each of these HIF target genes, and as such hold potential for both stratification of patients based on a LIMD1[low], SLC2A1[high]/IGF2BP2[high]/GAPDH[high] genetic profile and targeted therapies based upon this HIF target gene signature.

Taken together, our findings hold significant impact for the aetiology of *LIMD1*-negative lung cancers and hold the potential for advances in the diagnosis and prognosis of such cancers with respect to deregulated HIF regulation and associated oncogenic phenotypes, and subsequently hypoxia-targeted therapies.

# Materials and Methods

### Bioinformatic analysis

As a reference point for referring to positions within the *LIMD1* promoter, the unconfirmed transcriptional start site (TSS) was assigned according to the NCBI reference sequence NM_014240.2. This corresponds to nucleotide 45636323 on the primary chromosome 3 ref assembly NC_000003.11 and is 49-bp upstream from the AUG translation initiation codon. The human *LIMD1* promoter, which was preliminarily designated as 2.5-kbp upstream of the ATG translation initiation codon, was scrutinised using the Ensembl Genome Browser (http://www.genome.ucsc.edu) for the presence of CpG Islands, utilising the default software thresholds. The *in silico* screen for transcription factor binding motifs within the promoter was performed using the MatInspector software programme (http://genomatix.de) using the Matrix Family Library Version 8.1. HRE3

multiple sequence alignment was performed using *LIMD1* promoter sequences from the UCSC Genome Browser and aligned using Clustal Omega. HRE3 sequence logo was generated using WebLogo.

### Copy number analysis (CBioportal)

Provisional TCGA LUAD and LUSC datasets were accessed and downloaded from cbioportal.org. The datasets included 512 (LUAD) and 498 (LUSC) patients. Focally deleted genes altered in lung adenocarcinoma and squamous cell carcinoma were selected from a study by Campbell *et al* (2016) and used, along with LIMD1, to query LUAD and LUSC provisional datasets (via http://www.cbioportal.org). Putative copy number alterations from GISTIC were used to identify the following copy number categories: amplification, gain, diploid, shallow deletion and deep deletion for each gene. The percentage of samples displaying each category was then calculated for each gene. For each sample, linear LIMD1 copy number values were plotted against *LIMD1* mRNA expression values (log2-transformed RNA Seq V2 RSEM). Pearson's $r$ correlation analysis was performed using GraphPad Prism version 7.04.

### The Cancer Genome Atlas (TCGA) analysis

All analyses were conducted in the R statistical environment (v 3.3.2).

#### Data
Level-3 RNA-Seq data for human lung adenocarcinoma (LUAD) primary tumours ($n = 517$) were downloaded from the TCGA using the *TCGAbiolinks* R package (Cancer Genome Atlas Research Network, 2014; Colaprico *et al*, 2016).

#### Correlation
Pearson's product-moment correlation coefficient and $P$-value of pairwise comparisons between the mRNA abundance values of LIMD1 and hallmark hypoxia gene set (GSEA) were computed.

#### Survival
Patients were stratified into three risk groups based on the quartile values of mRNA abundance densities. The prognostic values of SLC2A1, GAPDH and IGF2BP2 in lower quartile ($n = 127$) and upper quartile ($n = 127$) risk groups were assessed using a Cox proportional hazards regression model, with $P$-values estimated using log-rank test. The survival modelling and Kaplan–Meier analyses were conducted using the *survival* package (v 2.41-3; Therneau & Grambsch, 2000). All analyses were conducted for a 5- and 10-year survival timeframe.

### Cell culture

U2OS, HEK293T, HeLa and A549 cells were maintained in D-MEM (Sigma) supplemented with 10% FCS and 1% pen/strep solution in a humidified 37°C incubator and 5% $CO_2$. SAEC were maintained in complete small airway epithelial cell growth medium (Lonza). Cells were regularly tested for mycoplasma. Hypoxic incubations were carried out at 1% $O_2$ within a humidified ProOx110 controller chamber (BioSpherix Ltd, New York, USA). Cells were transfected using Viafect (Promega E4981) with a 3:1 ratio of Viafect:DNA.

### Promoter cloning

Site-directed mutagenesis reactions were performed using Quik-Change XL Site-Directed Mutagenesis Kit (Stratagene #200517) as per the supplied protocol as confirmed by sequencing (Source Bioscience UK Limited, Nottingham). Primer sequences are supplied in Appendix Table S10.

### shRNA plasmids generation and transduction

shRNA sequences (supplied in Appendix Table S10) were annealed and ligated into psiRNA-DUO plasmid (Invivogen # ksirna4-gz3) with Acc65I and HindIII (NEB #R0599S and #R0104S).

The knockdown-rescue shRNA lentiviral system was a kind gift from Greg Longmore (Feng *et al*, 2010). The lentiviral system allows simultaneous shRNA-mediated knockdown of an endogenous target with concurrent rescue of the same RNAi-resistant target. Lentiviral plasmids containing a LIMD1 targeting shRNA with concurrent RNAi-resistant LIMD1 cDNA expression was modified so that expression of the cDNA was driven by the endogenous *LIMD1* promoter. A 2-kb region upstream of the LIMD1 ATG translation initiation codon was amplified using the primers ggagcgGTCGAC CAGGCACTTGGCATACAGATATGGTC (SalI forward) and cgctccGA ATTCGCTGCAGACAGGTGTCCGGGCCTAG (EcoR1 reverse). The ubiquitin C promoter from the pFlRu plasmid that drives the rescue expression was then replaced with the amplified *LIMD1* promoter. To create a rescue plasmid with a mutated HRE, site-directed mutagenesis with the already described ΔHRE3 primers was used. Lentivirally transduced cell lines were created as previously described (Foxler *et al*, 2012).

### Luciferase assays

The HRE-pGL3 luciferase construct containing three copies of the HRE from the phosphoglycerate kinase promoter was a kind gift from Thilo Hagen (Department of Biochemistry, National University of Singapore). The HRE element was subcloned into the pNL1.1 vector (Promega). Cells were co-transfected with 50 ng of HRE reporter vector and 5 ng firefly normalisation reporter plasmid DNA per well of a 12-well plate and lysed 36 h post-transfection in 1× Passive Lysis Buffer (Promega E1941). Luciferase activity assayed using a Nano-Glo Dual-Luciferase Reporter Assay System (Promega N1610). The 3′UTR sequences for HIF-1α and HIF-2α were PCR cloned from a HeLa cell cDNA and into the psiCheck2 vector using 5′XhoI and 3′NotI sites incorporated into the PCR primers. Primer sequences are supplied in Appendix Table S10.

### qRT–PCR

All qRT–PCR was performed using a 1-step RT–qPCR method (Promega A6020). All reactions were performed in triplicate with 20 ng of RNA 200 nM forward and reverse primers in a 25 μl reaction volume run on an ABI7000 instrument (Applied Biosystems). Gene-specific primers were designed to span an exon boundary and data normalised to the housekeeping genes RNA polymerase II. RNA extractions from cells were performed using a column-based purification (Promega Z6011) and from xenograft tissue following homogenisation in Tripure (Roche Applied Science 11667157001)

and aqueous phase extraction. The list of primers used appear in Appendix Table S10.

### VEGF-A ELISA

ELISAs were performed on cell supernatants following 48-h hypoxia utilising the Human VEGF-A Quantikine ELISA Kit (R&D Systems DVE00).

### Chromatin immunoprecipitation (ChIP)

$1 \times 10^7$ cells were stimulated overnight in hypoxia. Formaldehyde (1% v/v) was added to cross-link protein–DNA for 10 min at 37°C and was quenched with ice-cold 0.125 M glycine/PBS. Cells were harvested in 1 ml harvesting buffer (0.125 M glycine, 1 mM EDTA and protease inhibitors in PBS) and pelleted by centrifugation at 3,500 $g$ for 10 min at 4°C. The cell pellet was resuspended in 100 μl of lysis buffer (50 mM Tris–HCl pH8.0, 1% SDS, 10 mM EDTA plus protease inhibitors) and incubated on ice for 10 min. 50 μl of dilution buffer (20 mM Tris–HCl pH8.0, 1% Triton X-100, 2 mM EDTA, 150 mM NaCl plus protease inhibitors) was added and lysates sonicated on ice to shear the DNA to 200–600 bp. Lysates were cleared of insoluble material by centrifugation at 13,000 $g$ for 10 min at 4°C. An input sample was taken, and the remaining soluble chromatin containing supernatant diluted to 1 ml with dilution buffer and added to an antibody-conjugated IP matrix and incubated overnight at 4°C with rotation. IP matrix beads were washed 6 × 1 ml RIPA and protein–DNA complexes eluted in 2 × 75 μl elution buffer (1% SDS, 0.1 M NaHCO₃) for 15 min at room temperature with rotation. Cross-links were then reversed for 6 h at 65°C with NaCl (0.2 M), followed by incubation with 20 μg proteinase K, 40 mM Tris–HCl pH 6.5, 10 mM EDTA. DNA was then purified using the Qiagen PCR purification kit.

### Electrophoretic mobility shift assay

Electrophoretic mobility shift assays were performed using the method described in Foxler *et al* (2011) following incubation of cells at 1% O₂ for 24 h, prior to nuclear extraction. MG132 was included in all buffers to prevent degradation of HIF-1α. 5 μg of nuclear extracts was incubated in a total volume of 20 μl HIF binding buffer (50 mM KCl, 10 mM Tris pH 7.7, 5 mM DTT, 1 mM EDTA, 2 mM MgCl₂, 5% glycerol, 0.03% NP-40 and 400 ng salmon testes DNA) with 1 μl of HIF-1α or HIF-2α antibodies. Binding reactions were pre-incubated on ice for 30 min prior to addition of ³²P-labelled probe with an overnight incubation at 4°C. Complexes were then resolved on 5% polyacrylamide gels (acrylamide: bisacrylamide 30:0.8) at room temperature. Gels were dried and developed using a Fuji-film LAS-3000 phosphor-imager. For specificity of the HIF binding site and for competition assays, probes with the HIF site mutated were also used. Probes were synthesised by Sigma-Aldrich and listed in Appendix Table S10.

### Immunoblotting

Protein lysates were analysed using SDS–PAGE using standard Western blotting protocols. The list of antibodies used appear in Appendix Table S11.

### Cycloheximide (Cx) treatment

Cells were plated 24 h prior to treatment with cycloheximide (200 μg/ml; sc-3508). Media containing 400 μg/ml Cx were incubated in 1% O₂ for 24 h to allow de-oxygenation prior to addition to an equal volume of cell media. Lysates were taken immediately prior to Cx addition (start-point) and following the last time-point without Cx added (endpoint) to disseminate between drug-induced and endogenous HIF-α protein turnover. Cx was added so that all drug treatment times finished at the same time. HIF-α protein expression was assayed by Western blot and quantified with ImageJ software and normalised to β-actin loading control. The rate of turnover was calculated from the gradient of log[HIF-α protein] against time.

### Animal studies

All animal experiments conformed to the British Home Office Regulations (Animal Scientific Procedures Act 1986; Project License PPL 70/7263 to Prof Nick Lemoine). Trial experiments and experiments done previously were used to determine sample size with adequate statistical power. Twenty-five mice for each group were studied in total in two independent experiments. Eight- to 12-week-old female SCID/beige mice (Harlan Laboratories) were given a subcutaneous injection of $5 \times 10^6$ transduced A549 cells in 100 μl PBS into the right flank for subcutaneous tumour growth. Calliper measurements were taken over time, and the experiment reached an endpoint when the first tumour measurement of maximum length x maximum breadth exceeded the maximum size dictated by the Project licence. Mice were killed by Schedule 1 cervical dislocation. Xenografts were then immediately excised prior to the onset of rigour mortis. Xenografts were bisected and half flash frozen in liquid N₂ or frozen on dry ice in 0.8 ml Tripure for subsequent RNA extractions. Snap-freezing of fresh subcutaneous tumours is the best recognised method for subsequent effective blood vessel immunostaining. This method avoids many of the limitations of prefixing the tissue that can actually reduce antigenicity when it comes to immunostaining for blood vessels.

### Xenograft analysis

Snap-frozen xenograft samples were sectioned into 4–6 μM sections (Pathology Department, Bart's Cancer Institute, Queen Mary University of London). Sections were fixed in ice-cold acetone for 10 min and stained overnight with the endothelial blood vessel marker endomucin (1/100, Santa Cruz V7C7) and AlexaFlour 546-conjugated secondary antibody and mounted in Prolong Gold anti-fade reagent with DAPI (Invitrogen, Paisley, UK). Stained sections were visualised on a Zeiss Axioplan microscope, and blood vessel number and tumour section area systematically counted and measured for the whole section area. For each xenograft, blood vessel density was calculated across a midline tumour section using the formula (Σ number of blood vessels)/(Σ section area). RNA was extracted using the manufacturer's recommended protocol following homogenisation in Tripure reagent.

      

### *In vitro* clonogenic and proliferation assays

A549 HRE wild-type and mutant cells were seeded into 6-cm plates at $5 \times 10^4$ cells per plate. Twenty-four hours after seeding plates were placed into 1% $O_2$. Forty-eight hours postseeding, three plates for each condition were trypsinised and counted using a TC20TM BioRad Cell Counter. Each plate was counted in duplicate and an average cell count calculated. Plates were counted 48, 72, 120 and 168 h postseeding, and growth curves generated. Four biological repeats of this experiment were conducted.

### Microarray analysis

RNA was extracted from HBEC treated for 72 h with 80 nM scrambled or LIMD1 targeting siRNA in quadruplet. Microarray analysis was performed by the Genome Centre, Bart's Cancer Institute using a HT12v4.0 Illumina array. Raw data were normalised and fold-change gene expression calculated from the average expression value for each condition. Genes with a q value cut-off of < 0.15 were analysed by ingenuity pathway analysis (IPA) [Qiagen] software using fold-change values for each mRNA: siRNA LIMD1 versus siRNA scrambled. Gene ontology was collected from the Bio Functions read-out of IPA results where activation was > +1 or < −1 (Dataset EV1), and categories were collapsed into similar overall functions, e.g. apoptosis and cell survival, and directionality of function was inferred from the activation z-score in Dataset EV1. The HIF-1α transcription factor was indicated as an activated upstream regulator by IPA (HBEC activation score 3.114 and P-value of overlap 1.76E-04); therefore, this was also represented in the heat map. These categories were then applied to the heat map for functional clustering; only genes that were placed in these enriched categories are shown. IPA upstream analysis also produced a network of HIF-1α interactions for HBEC siRNA LIMD1 versus siRNA scrambled treated. These interactions and the predicted activation state of downstream effectors are inferred from the total gene set submitted for IPA (q < 0.15).

### Patient cohort and immunohistochemistry

The overall number of patients within the cohort was 276 of which 276 were valid for immunohistochemistry staining. The cohort consisted of 150 males and 126 females with an age range of 36–91 years. Ninety-three percentage of cases were adenocarcinoma, 3% small-cell carcinoma and 4% other types, and all 276 tumour cores were of primary lung tumour origin. Informed consent was obtained from all subjects, and the experiments conformed to the principles set out in the WMA Declaration of Helsinki and the Department of Health and Human Services Belmont Report. IHC was carried out as previously described (Sharp *et al*, 2008; Spendlove *et al*, 2008). Briefly, slides were heated to 60°C on a hotplate (Leica, HI1220) for 10 min, allowed to cool for 5 min before being dewaxed in a Leica autostainer. Antigen retrieval was then performed via microwave for 20 min in citrate buffer pH 6.0 for VEGF sections. Antigen retrieval for LIMD1 was carried out in a water bath Epitope retrieval solution 2 pH 9.0 (Bond) at 95°C for 35 min. Slides were peroxidase treated (Novolink) for 5 min in Sequenza trays, washed with TBS twice for 5 min and blocked (Novolink) for 5 min. The working dilutions of the antibodies were then made up in antibody diluent (Bond leica); LIMD1 1/200, and

**The paper explained**

**Problem**

Within solid tumours, including lung cancer, inadequate oxygen levels (hypoxia) create an environment that is a driving force of cancer progression and form a resistance mechanism to all forms of therapy such as standard chemotherapies and ionising radiation. In chronic hypoxia, the transcriptional regulators of hypoxia (HIFs) are degraded through negative feedback loops; however, neoplastic cells evade this to survive in this harsh microenvironment. How this occurs in non-small-cell lung cancer is poorly understood and serves as an area of significant interest for cancer biology and potential hypoxia-based targeted therapies.

**Results**

We identified that the tumour suppressor LIMD1, which facilitates efficient degradation of the transcriptional regulator of hypoxia (Foxler *et al*, 2012), is itself a HIF target gene. This creates a negative feedback loop whereby the activity of HIF is limited under prolonged hypoxic exposure, and mitigates pro-tumorigenic effects of hypoxia. Subcutaneous implantation of cells lacking this feedback mechanism formed larger and more vasculature *in vitro*. Furthermore, *in silico* analysis of TCGA data shows that LIMD1 is lost in 47% of lung adenocarcinoma and serves as an independent prognostic marker. Deeper analysis of this dataset reveals a negative correlation between LIMD1 and a hypoxic gene set which further correlates with patient outcome.

To conclude, we have identified a novel LIMD1-mediated negative feedback loop of HIF regulation that effects tumour growth, highlighting the functional importance of *LIMD1* expression in normal lung homeostasis and the tumorigenic advantage its loss/deregulation gives to the hypoxic tumour microenvironment.

**Impact**

Our findings open a new field of research into the aetiology, diagnosis and prognosis of *LIMD1*-negative lung cancers and hold the potential for advances in the stratification of patients with respect to deregulated HIF regulation and associated phenotypes. This holds the potential for development of chemotherapeutic drugs that target *LIMD1* loss which could be used in combination with hypoxia-targeted therapies.

VEGF-A Pre-diluted (SP28, Abcam). Beta-2-microglobulin 1/2,000 (Dako, A0072) was used as a reference positive control, and negative controls were without primary antibody. Each antibody at its chosen dilution was incubated for 60 min before being washed (×2). Secondary and tertiary reagents (Novolink postprimary and polymer) were incubated on the slides for 30 min each with a washing step between. Slides were developed with DAB solution (Novolink) and counterstained with haematoxylin (Novolink) for 6 min. Slides were then dehydrated and mounted in DPX prior to observation and laser scanning.

### Statistics

Statistical analyses were performed using R 3.4.4 and are described in each figure legend. Where systematic differences existed between experimental runs, data were analysed with mixed-model ANOVA allowing a separate intercept and effect of time for each run. Homoscedasticity and normality of all model residuals were evaluated graphically. Where residuals were non-normal or heteroscedastic, the model was refit to the $\log_{10}$-transformed dependent variable. If model assumptions were still not met, nonparametric tests or

Welch-corrected Student's *t*-tests were used instead. Where data are normalised to a group, this group was excluded from the analysis. One-sample Student's *t*-tests were used when comparing data to a standardised group (theoretical value of 1).

## Data availability

Microarray data from this publication have been deposited to Gene Expression Omnibus and assigned the identifier accession number GSE114692.

**Expanded View** for this article is available online.

## Acknowledgements

This work was supported by RCUK funding awarded to TVS from the BBSRC (grant BB/L027755/1), MRC (grant MR/N009185/1) and CRUK (grant CRUK-A12733). PR is funded by a joint Royal College of Surgeons/Cancer Research UK Clinician Scientist Fellowship in Surgery (C19198/A15339). PR and JGF are both supported by the Barts Charity and the Orchid Charity (0000 0003 1861 7984) (to NRL). MJP is a BBSRC New Investigator (BB/N018818/1). DL is a Medical Research Council New Investigator Research Grant holder (MR/L008505/1).

## Author contributions

DEF, KSB, JGF, PG, SC, KMS, KMD, AN, EG, HIR, PTK, MAH, T-YC, PES, LER and TVS designed and performed experiments and analysed the data. TRM, H-WW, PSR, MJP, DL, NRL, PR, TAG, CC, KMH-D and IS provided reagents, experimental advice and design. All authors contributed to editing and proofreading the manuscript. DEF, KSB and TVS wrote the manuscript. TVS supervised and managed all research.

## Conflict of interest

The authors declare that they have no conflict of interest.

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
