## [Review Process File · EMBO Molecular Medicine]

A HIF-LIMD1 negative feedback mechanism mitigates the pro-tumorigenic effects of hypoxia

Daniel E. Foxler, Katherine S. Bridge, John G. Foster, Paul Grevitt, Sean Curry, Kunal M. Shah, Kathryn M. Davidson, Ai Nagano, Emanuela Gadaleta, Hefin I. Rhys, Paul T. Kennedy, Miguel A. Hermida, Ting-Yu Chang, Peter E. Shaw, Louise E. Reynolds, Tristan R. McKay, Hsei-Wei Wang, Paulo S. Ribeiro, Michael J. Plevin, Dimitris Lagos, Nicholas R. Lemoine, Prabhakar Rajan, Trevor A. Graham, Claude Chelala, Kairbaan M. Hodivala-Dilke, Ian Spendlove, Tyson V. Sharp

Review timeline:

Submission date:	26 July 2017
Editorial Decision:	01 September 2017
Revision received:	30 March 2018
Editorial Decision:	15 May 2018
Revision received:	23 May 2018
Accepted:	25 May 2018

Editors: Roberto Buccione & Céline Carret

Transaction Report:

1st Editorial Decision

01 September 2017

Thank you for the submission of your manuscript to EMBO Molecular Medicine. We are very sorry that it has taken so long to get back to you on your manuscript.

In this case we experienced unusual difficulties in securing three willing and appropriate reviewers due in part to the overlap with the holiday season. As a further delay cannot be justified I have decided to proceed based on the two available consistent evaluations.

As you will see, although both Reviewers find the study of interest, they express a number of concerns. I will not go into much detail as the comments are in general clear and detailed. However, I would like to emphasise that the important technical and quality issues mentioned by reviewer 1 do hamper interpretation as can be inferred from reviewer 2's comments.

While publication of the paper cannot be considered at this stage, we would be pleased to consider a revised submission, with the understanding that the both Reviewers' concerns must be fully addressed including with additional experimental data where appropriate and that acceptance of the manuscript will entail a second round of review.

Please note that it is EMBO Molecular Medicine policy to allow a single round of revision only and that, therefore, acceptance or rejection of the manuscript will depend on the completeness of your responses included in the next, final version of the manuscript.

As you know, EMBO Molecular Medicine has a "scooping protection" policy, whereby similar findings that are published by others during review or revision are not a criterion for rejection. However, I do ask you to get in touch with us after three months if you have not completed your revision, to update us on the status. Please also contact us as soon as possible if similar work is published elsewhere.

Please note that EMBO Molecular Medicine now requires a complete author checklist (<http://embomolmed.embopress.org/authorguide#editorial3>) to be submitted with all revised manuscripts. Provision of the author checklist is mandatory at revision stage; the checklist is designed to enhance and standardize reporting of key information in research papers and to support reanalysis and repetition of experiments by the community. The list covers key information for figure panels and captions and focuses on statistics (please note reviewer 1's comments on this aspect), the reporting of reagents, animal models and human subject-derived data, as well as guidance to optimize data accessibility. In this case, the author checklist is especially relevant as, in addition to the concerns on the clinical features of the TMA, I note that both reviewers have reservations on your presentation of statistics information. The Author checklist will be published alongside the paper, in case of acceptance, within the transparent review process file.

We now mandate that all corresponding authors list an ORCID digital identifier. You may acquire one through our web platform upon submission and the procedure takes <90 seconds to complete. We also encourage co-authors to supply an ORCID identifier, which will be linked to their name for unambiguous name identification.

Please carefully adhere to our guidelines for authors (<http://embomolmed.embopress.org/authorguide>) to accelerate manuscript processing in case of acceptance.

I look forward to seeing a revised form of your manuscript in due time

***** Reviewer's comments *****

Referee #1 (Comments on Novelty/Model System for Author):

I would have graded the technical quality as overall high with the exception of the statistical analysis, the xenograft data, and the quality of the TMA IHC.

Referee #1 (Remarks for Author):

Summary. The authors have a considerable track record in investigation and discovery of LIMD1 functions in cancer and hypoxia. Here, they extend their work to investigate and ultimately elucidate HIF-1 mediated control of LIMD1 transcription and hypoxic negative feedback of LIMD1 on HIF-1 α protein expression/stability.

General comments. Overall the discovery of a new mechanism for regulation of HIF-1 α stability in hypoxia is of considerable impact. However, the significance of this finding demands that the authors test whether this is a general finding in mammalian cells, or only associated with malignancy. As such, the authors need to perform key, selected experiments in non-tumorigenic cell lines or even better, primary human or mouse embryonic fibroblasts to discern functions in normal cell biology. At present the differences in gene expression and protein level changes are relatively modest across the work. Presumably this is due to background signaling cross talk in malignant cells. Thus, it is possible that the signal to noise could be increased in normal versus malignant cells.

The data from the studies in intact mice and patient samples are more problematic. The variance is huge both for xenograft tumor size and blood vessel density. The immunofluorescence suggests incomplete microvessel detection. There are two likely reasons for this. First, the harvest and preparation of the tumor tissue is inadequate. Most labs perfuse the mice with formalin via the left ventricle to rapidly and evenly distribute the fixative throughout the tumor tissue. This is followed by thin tissue slicing and overnight fixation. Second, molecular microvessel expression in tumors is heterogeneous and endomucin in particular does not detect all vessels. As such, "cocktails" of CD31/endomucin or CD31/VE-cadherin should be performed for optimal vessel detection/enumeration. Finally, the data variance for HIF target gene expression in tumors is also huge and not normally distributed. Non-parametric statistical tests need to be done here as it was for tumor size. For the patient TMAs, Figures D and E are confusing, but suggest that LIMD1 is upregulated in most tumors given the "frequency" label for the Y-axis. However, this frequency of upregulation does not support the data suggesting that LIMD1 expression is lost in lung cancers.

The data would be clearer if presented as number of cores positive for each expression level/category. The IHC analysis and the tissue morphology should also be improved. Currently, the morphology is "fuzzy" will cellular structures ill-defined, and the IHC is overdeveloped such that it is difficult to discern nucleus from cytoplasm which is important as presumably LIMD1 should be nuclear localized for interaction with hypoxia-induced HIF-1alpha. Overall, the association of LIMD1 and VEGF expression with prognosis is statistically borderline, such that inferences from these data should be quite cautionary until independent studies are performed or larger patient cohorts analyzed.

Finally, I am concerned about the use of multiple t-tests in Figures 1-5. One way or in two parameter instances (time) two-way ANOVA is more appropriate. Beyond GraphPad a statistician should be engaged for these data.

Referee #2 (Remarks for Author):

This is a technically sound study that shows that LIMD1, an adaptor protein involved in the downregulation of HIF transcription factors is regulated by hypoxia and is a direct transcriptional target of HIFs. This suggests a negative feedback loop in hypoxic signaling via upregulation of LIMD1. Moreover hypoxia-induced expression of LIMD1 reduces tumor growth of xenotransplanted tumor cells, and LIMD1 correlates with a better prognosis of lung tumor cases. These are interesting findings which expand previous studies from the group and support an role of LIMD1 in hypoxic signaling in cancer. However there are some specific questions/problems that need to be clarified before publication.

Fig. 1 E: It seems more meaningful to show data of Fig. S1G here instead of the deletion analysis because these make the point more clearly by showing specific point mutants of HREs.

Fig. 2D Why is LIMD1 not induced under hypoxia in this experiment?

Fig. 3B-D needs better description and interpretation: There is a strong drop in HIF1a from 4 hours to 24 hours of hypoxia which is not commented by the authors. Only a minor portion of this drop can be related to exogenous HRE-driven LIMD1 expression because HIF1a is still much lower in the HREmut cells at 24 hrs than at 4 hrs. In line the differences in HIF reporter activity and target gene expression between HERwt and HERmut cells are relatively small, raising the question of their significance. The authors should address this issue and also use shorter time points to capture the optimal time window for LIMD1 activity. Is the massive decline in HIF1a between 4 and 24 hrs due to endogenous-LIMD1 which is not efficiently depleted by the shLIMD1?

Why are differences between flag bands from HREwt and HREmut much smaller than between LIMD1 bands if both blots detect the same (exogenous) protein? Again, what is the contribution of endogenous LIMD1 in this setting and how well did the shRNA to LIMD1 perform in these cells? Related to that, in Fig. S3B it appears that there is no specific increase of LIMD1 in the HREwt transduced cells under hypoxia, rather there seems to be a decrease in the HREmut transduced cells. How can the authors explain this?

Fig. 7 Please explicitly state whether correlations between LIMD1 and HIF1 or LiMD1 and VEGF expression are positive or negative. It is not possible to extract this information from Fig. S7.

The discussion section might be shortened; in particular with respect to parts dealing with VEGF targeted therapies where the connection to the presented data is rather loose.

1st Revision - authors' response

30 March 2018

I) General comments. Overall the discovery of a new mechanism for regulation of HIF-1alpha stability in hypoxia is of considerable impact. However, the significance of this finding demands that the authors test whether this is a general finding in mammalian cells, or only associated with malignancy. As such, the authors need to perform key, selected experiments in non-tumorigenic cell

lines or even better, primary human or mouse embryonic fibroblasts to discern functions in normal cell biology.

We thank the reviewer for these excellent suggestions. As requested, we have extended our observations to including the non-transformed primary human small airway epithelial cells (SAEC) and primary human dermal fibroblasts (hDF). In both these cell types we have also observed the new proposed HIF-LIMD1 regulatory mechanism to exist

The key experiments we performed using these new cultures/lines were as follows:

*Using the SAEC and hDFs we performed a time-course of exposure to 1% O₂ demonstrating, as predicted by our model, that LIMD1 expression is induced over time in hypoxia in both SAEC and hDF, and this can be detected at both the mRNA and protein level (**Figure 1A, S1A, S1B**). SAEC are the non-transformed precursor for a subset of lung adenocarcinoma (1, 2); we therefore chose this cell line to perform subsequent analysis of the HIF-LIMD1 regulatory mechanism to compare to the cancer cell lines (U2OS, HeLa, A549). In the SAEC line, LIMD1 hypoxic induction was also dependent on HIF-1 α , as hypoxic induction of LIMD1 was prevented upon siRNA-targeted HIF-1 α knock-down (**Figure S2D**).*

*Next, we generated HRE-WT and HRE-MUT LIMD1-promoter cell lines in the SAEC, and observed that the hypoxic induction of LIMD1 under chronic hypoxic conditions (24h and 48h) was impaired in the HRE-MUT cell line (**Figure 3B, E, H and S3E**), as previously observed in U2OS and HeLa HRE-WT and HRE-MUT isogenic pairs. Furthermore, HIF-1 α turnover observed at these later time-points in the HRE-WT SAEC line was significantly impaired in the HRE-mutant line, demonstrating again, as our model predicts, that hypoxic induction of LIMD1 facilitates HIF-1 α degradation in this primary isogenic pair of cell lines. Finally, analysis of HIF-target gene transcription in the SAEC HRE-WT and HRE-MUT lines demonstrated small but consistent up-regulation of expression of these genes in the HRE-MUT line compared to HRE-WT under hypoxic conditions, demonstrating the transcriptional (and likely physiological) significance of the HIF-LIMD1 negative feedback in these primary cells (**Figure S3K**).*

We also attempted multiple times (unsuccessfully) to obtain stable lentiviral transduced lines using primary hDF to perform the same analysis as above. However, we are pleased that the above siRNA and lentiviral transduced SAEC data is supportive of the HIF-LIMD1 negative feedback mechanism in a primary line, and that LIMD1 hypoxic induction is also observed in primary hDF. It is for this reason that we therefore focused on the SAEC lines for subsequent analyses, and also due to its relevance to lung cancer and the physiological focus of the paper.

At present the differences in gene expression and protein level changes are relatively modest across the work. Presumably this is due to background signalling cross talk in malignant cells. Thus, it is possible that the signal to noise could be increased in normal versus malignant cells.

*We did not observe that the hypoxic induction of LIMD1 and associated regulation of HIF-1 α had a greater significant difference in this non-transformed SAEC compared to the cancer cell lines previously assayed. In fact, the observed mechanism appears if anything subtler. Indeed this observation is not unexpected, as many cellular pathways regulated by HIF are significantly up-regulated in cancer cell lines compared to their derived normal primary cell lines (3). This is supported by comparison of HIF-target gene induction between A549 (lung adenocarcinoma cell line) and SAEC (non-transformed airway epithelial cell line), where a lesser fold-change of induction of HIF target genes (including BNIP3, ALDOC, PHD2 and LIMD1) upon exposure to hypoxia is observed in the primary SAEC line (**Response Letter Figure 1-D**). Corroboratively, we observed that the upstream regulation of these genes by the HIF-LIMD1 negative feedback axis in the SAEC HRE^{wt}/HRE^{mut} lines was also not as dramatic as their transformed counterparts (**Figure S3K**).*

Fig. 1A. LIMD1 mRNA expression in a panel of cell lines, including non-transformed small airway epithelial cells (SAEC) and primary human dermal fibroblasts (hDF) exposed to hypoxia (1% O₂) for up to 48 hours.

Fig. S1A-B. LIMD1 protein expression in SAEC and hDF exposed to 1% O₂ for up to 48 hours, and western blot analysis of LIMD1 protein expression.

Fig. S2D. LIMD1 protein expression in SAEC exposed to normoxic (20% O₂) or hypoxic (1% O₂) conditions and treated with the indicated siRNAs. LIMD1 induction under hypoxic conditions is impaired upon knock-down of HIF-1α and HIF-2α.

Fig. 3B. SAEC transduced with a lentiviral vector which simultaneously knocks-down LIMD1 via shRNA and re-expresses an RNAi resistant LIMD1 from the wild type LIMD1 promoter (HRE^{wt}) or one containing a mutated HRE (HRE^{mut}) and exposed to 1% O₂ for up to 48 hours. LIMD1 hypoxic induction is impaired in the HRE^{mut} line, and HIF-1α protein is stabilised in this line relative to HRE^{wt}.

Fig. 3E and H. Western blot analysis of LIMD1 and HIF-1α protein in (B).

Fig. S3E. Densitometric analysis of LIMD1 protein expression in SAEC HRE^{wt/mut} lines, mean ± SEM, n=3.

Fig. S3K. qRT-PCR analysis of the indicated HIF target gene expression in SAEC HRE^{wt} and HRE^{mut} lines exposed to 1% O₂.

As such, the authors need to perform key, selected experiments in non-tumorigenic cell lines or even better, primary human or mouse embryonic fibroblasts to discern functions in normal cell biology.

*The analysis of this mechanism in the mouse revealed that the pertinent conserved HRE was not present in this species (Response Letter Fig. 1E). Therefore as one would expect, no hypoxic induction of mouse *Limd1* was observed in the mouse fibroblast NIH3T3 cell line at either the mRNA or protein level (as a positive control for hypoxic induction, *Bnip3* was assayed) (Response Letter Fig. 1F-H). However, in the course of this study we have identified an HRE within the promoter of the *Limd1* paralogue *Wtip* in this species. This discovery is the focus of an independent study, and is therefore beyond the scope of this manuscript. We wish this paper to remain firmly focussed on the novel LIMD1-HIF mechanism in human disease/lung cancer.*

Response Figure 1: these data are not included in the manuscript

(A-D) qRT-PCR analysis of the indicated HIF target gene expression in A549 and SAEC exposed to 20% O₂ or 1% O₂ for 24 hours. HIF target gene induction in response to hypoxic exposure is less marked in the non-transformed primary SAEC compared to a cancer adenocarcinoma counterpart.

(E) Sequence alignment of the LIMD1 promoter in humans and mice shows that the human HRE (CGTGG) is not conserved in the mouse promoter.

(F) Mouse *Limd1* mRNA expression in murine fibroblast cell line NIH3T3 exposed to 1% O₂ for up to 48 hours

(G) Western blot analysis of LIMD1 protein expression in NIH3T3.

(H) HIF target gene *Bnip3* mRNA was assayed as a positive control for hypoxic induction.

The text for all of the above has been changed and amended accordingly (highlighted in blue text) (page 5, lines 11-14 and page 6 line 17):

‘We therefore assessed endogenous *LIMD1* expression in a panel of cell lines exposed to 1% O₂ (henceforth referred to as hypoxia), including transformed/immortalised lines (A549, HeLa, HEK293 and U2OS), non-transformed small airway epithelial cells (SAEC) and primary human dermal fibroblasts (hDF).’

‘siRNA-mediated depletion of HIF-1α reduced LIMD1 protein and mRNA expression under hypoxic, and to a lesser extent, normoxic conditions in all cell lines examined.’

In summary to this reviewer’s important points, and additional experiments requested, we have performed these assays and can confidently conclude that this is a fundamental mechanism in both

normal and cancerous human cell types. We agree that it is therefore, as the reviewers kindly noted, 'of considerable impact' to this field of hypoxia biology and related disease.

2) The data from the studies in intact mice and patient samples are more problematic. The variance is huge both for xenograph tumor size and blood vessel density. The immunofluorescence suggests incomplete microvessel detection. There are two likely reasons for this. First, the harvest and preparation of the tumor tissue is inadequate. Most labs perfuse the mice with formalin via the left ventricle to rapidly and evenly distribute the fixative throughout the tumor tissue. This is followed by thin tissue slicing and overnight fixation. Second, molecular microvessel expression in tumors is heterogeneous and endomucin in particular does not detect all vessels. As such, "cocktails" of CD31/endomucin or CD31/VE-cadherin should be performed for optimal vessel detection/enumeration

*We would like to clarify here that the method used to prepare the tissue **did not include tissue fixation** because this is not appropriate for blood vessel detection in subcutaneous tumours. Indeed, in the Methods we stated that the tumours were snap frozen and then 5µm sections made. The reviewer is correct to say that for some normal tissues or for very large slow-growing tumours prefixation may be necessary, especially for transmission electron microscopy. However this is not the case here. Snap freezing subcutaneously grown tumours in mice is standard and best practice. This method means taking fresh tissue and snap freezing without fixation, and avoids the complications of fixing whole tumours using chemicals. Instead, 5µm sections are fixed prior to staining. We have expanded on this to make it clearer in the Methods in the revised manuscript. It is also worth noting that we published in Reynolds et al., Nature Medicine 2002 that, in this type of subcutaneous tumour, CD31 and VECAD staining give the same number of vessels in vessel counts. Indeed, endomucin, as we have used here, is now the gold standard for blood vessel staining in these fast growing tumours. Thus, given the fast growing nature of these tumours and the fact that we have used endomucin to detect the blood vessel we believe that the data provided gives the best indication of blood vessel numbers.*

In the revised Methods we have now added the following to clarify this important point:

'Snap-freezing of fresh subcutaneous tumours is the best-recognized method for subsequent effective blood vessel immunostaining. This method avoids many of the limitations of prefixing the tissue that can actually reduce antigenicity when it comes to immunostaining for blood vessels.' Page 23, lines 15-19.

3) Finally, the data variance for HIF target gene expression in tumors is also huge and not normally distributed. Non-parametric statistical tests need to be done here as it was for tumor size.

We agree with the reviewer and have now redone all statistical analysis within the study as requested, in collaboration with Dr Hefin Rhys (The Francis Crick Institute). These analyses replace those performed originally. Details of all statistical analyses and corresponding R files are contained within the Statistics Report submitted and the revised Methods (Page 26, lines 9 – 19):

'Statistical analyses were performed using R 3.4.4 and are described in each figure legend. Where systematic differences existed between experimental runs, data were analysed with mixed model ANOVA allowing a separate intercept and effect of time for each run. Homoscedasticity and normality of all model residuals were evaluated graphically. Where residuals were non-normal or heteroscedastic, the model was re-fit to the log₁₀-transformed dependent variable. If model assumptions were still not met, non-parametric tests or Welch-corrected Student's t tests were used instead. Where data are normalised to a group, this group was excluded from the analysis. One-sample Student's t tests were used when comparing data to a standardised group (theoretical value of 1).'

Welch's t-test for non-normal distribution upon log₁₀ transformed mRNA values was performed for the A549 WT and MUT HRE xerographs.

4) For the patient TMAs, Figures D and E are confusing, but suggest that LIMD1 is upregulated in most tumors given the "frequency" label for the Y-axis.

We would like to clarify that this immunohistochemical staining and scoring was conducted upon a cohort of lung adenocarcinoma and squamous cell carcinoma patient samples and is not compared to a 'normal' sample e.g. matched adjacent tissue. The observed LIMD1 expression in these samples cannot therefore be up- or down-regulated as it is not a comparative analysis to 'normal'. These data demonstrate the range of expression of LIMD1 within the tumour samples, which was then correlated to overall survival of these patients.

However, this frequency of upregulation does not support the data suggesting that LIMD1 expression is lost in lung cancers. The data would be clearer if presented as number of cores positive for each expression level/category.

This is not what the data suggests statistically. Our analysis reveals that tumours with LIMD1 expression lower than the median for the cohort are associated with significantly worse survival ($p=0.045$). Conversely, patient survival is worse for those whose tumours express higher than median levels of VEGF within the cohort ($p=0.049$).

5) The IHC analysis and the tissue morphology should also be improved. Currently, the morphology is "fuzzy" with cellular structures ill-defined, and the IHC is overdeveloped such that it is difficult to discern nucleus from cytoplasm which is important as presumably LIMD1 should be nuclear localized for interaction with hypoxia-induced HIF-1 α .

We agree with the reviewer and apologise for the mistake of inclusion of these low resolution images. We now include the correct resolution version of these images (Fig. 6A and S6A).

Both oxygen labile HIF subunits and their regulatory proteins (including LIMD1, PHD2 and VHL) are found localised within the nuclear and cytoplasmic compartments of the cell, and degradation of HIF- α has been demonstrated to occur in both compartments (4-11). Immunohistochemical staining of LIMD1 expression, both within this study and as published, agrees with these findings (9, 12). We thank the reviewer for highlighting this point, which has been addressed in the manuscript text (Page 11, lines 2-4):

'In agreement with previous studies, LIMD1 protein expression was detected in both nuclear and cytoplasmic compartments.'

6) Overall, the association of LIMD1 and VEGF expression with prognosis is statistically borderline, such that inferences from these data should be quite cautionary until independent studies are performed or larger patient cohorts analyzed.

We agree with the reviewer and have reduced the inference from the immunohistochemistry performed on this cohort, which contains a relatively small number of patients. In addition, we have now interrogated The Cancer Genome Atlas Lung Adenocarcinoma (TCGA-LUAD) data collection in order to determine the contribution of LIMD1 and a larger panel of HIF target genes on patient survival.

Firstly, copy number analysis of LIMD1 in lung adenocarcinoma ($n=512$) and lung squamous cell carcinoma ($n=498$) cohorts demonstrated that combined single allele (shallow) or bi-allele (13) deletion of the LIMD1 gene occurred in 47.1% and 85.4% of patients, respectively (Figure 6D and S6D).

Secondly, regression analysis demonstrated correlation between LIMD1 copy number and mRNA expression (Figure S6E-F), therefore patients were stratified into risk groups (quartiles) based on mRNA abundance intensities and patient survival was determined using a Cox proportional hazards model. This analysis determined that patients in the risk group exhibiting high LIMD1 expression (High Exp) had increased overall survival, whereas patients exhibiting low LIMD1 expression (Low Exp) had reduced overall survival (log rank $P = 0.021$, HR 0.6) (Figure 6E). This data significantly corroborates the survival data based on IHC analysis performed in our small IHC cohort study. Of note, LIMD1 expression with the lung squamous cell carcinoma cohort did not significantly affect overall survival (Response Figure 2A). However, it is interesting to note that the degree of LIMD1 copy number alteration in SCC was over 85.4% of tumour samples. Why this does not correlate with

poor survival in SCC is unknown, and warrants further investigation, which is beyond the scope of this current study.

*Kaplan-Meier analysis of VEGFA or -B determined no significant association between expression of these factors and survival (**Response Figure 2B-C**). Furthermore, a number of studies have identified VEGF expression to be uninformative as a hypoxic-marker with regards to cancer disease progression or patient outcome due to its regulation by non-hypoxic stimuli and its secretion (14-16). This is particularly pertinent in non-small cell lung cancer which, in addition to contending with these non-HIF-regulatory factors, is also derived from a highly vascularised tissue. Therefore, due to the disreputability of VEGF as a hypoxic-marker, we analysed the TCGA LUAD cohort to identify hypoxia/HIF signature genes correlated with LIMD1 expression, and then determined survival of patients based on expression of these HIF target genes.*

*We identified a strong inverse correlation between LIMD1 expression and HIF target genes GLUT-1 (SLC2A1), GAPDH and IGF2BP2 (**Figure 6F-H**). The mean mRNA abundance of each gene was highest in the LIMD1 Q1 group and lowest in the LIMD1 Q3 group. Kaplan-Meier analysis of survival of patients stratified by expression of these genes revealed that patients with the highest expression of these genes have significantly worse overall survival compared to patients who demonstrate the lowest expression (**Figure 6I-K**). Conversely, the HIF target gene ENO2 was not correlated with LIMD1 expression in the lung adenocarcinoma cohort, and there was no significant association between expression of ENO2 and patient survival (**Response Figure 2D-E**).*

The new TCGA analysis is described in the manuscript text (Page 11, line 11 – Page 12, line 6).

Fig. 6D. Copy number alterations of *LIMD1* and other validated tumour suppressor genes from a lung adenocarcinoma (LUAD) and **Fig. S6D** lung squamous cell carcinoma (LUSC) TCGA cohort.

Fig. S6E-F. Regression analysis performed upon the above lung adenocarcinoma and squamous cell carcinoma cohort from TCGA demonstrates significant correlation between *LIMD1* copy number and mRNA expression.

Fig. 6E. Stratification of TCGA LUAD based on *LIMD1* mRNA expression demonstrates worse overall survival for patients within the lowest *LIMD1*-expressing quartile (Low Exp) compared to the highest *LIMD1*-expressing quartile (High Exp) within the LUAD cohort.

Fig. 6F-H. Correlation analysis of *LIMD1* mRNA expression in TCGA LUAD cohort identified a significant inverse correlation between *LIMD1* and HIF target genes *SLC2A1*, **(D)** *GAPDH* and **(E)** *IGF2BP2*. Q1 = lowest *LIMD1* mRNA expression quartile, Q3 = highest *LIMD1* mRNA expression quartile. The mean mRNA abundance of each gene is highest in the *LIMD1* Q1 group and lowest in the *LIMD1* Q3 group.

Fig. 6I-K. Stratification of TCGA LUAD cohort into quartiles based on *SLC2A1*, **(G)** *GAPDH* and **(H)** *IGF2BP2* demonstrates worse overall survival for patients within the highest expressing quartile (High Exp) compared to the lowest expressing quartile (Low Exp) for each gene.

**Response Figure 2: These data are not included in the manuscript**

(A) Stratification of TCGA lung squamous cell carcinoma (LUSC) based on LIMD1 mRNA expression demonstrates no significant difference upon overall survival between lowest LIMD1-expressing quartile (Low Exp) compared to the highest LIMD1-expressing quartile (High Exp).

(B) Stratification of TCGA lung adenocarcinoma (LUAD) cohort into quartiles based on VEGF-A and (C) VEGF-B mRNA expression demonstrates no significant effect of expression of either VEGF isoform upon overall survival.

(D) LIMD1 mRNA expression is not correlated to expression of HIF target gene ENO1 within TCGA LUAD cohort.

(E) ENO1 expression demonstrates no significant effect upon overall patient survival in LUAD.

7) Finally, I am concerned about the use of multiple t-tests in Figures 1-5. One way or in two parameter instances (time) two-way ANOVA is more appropriate. Beyond GraphPad a statistician should be engaged for these data.

We refer the reviewer to point 3 above where in collaboration with Dr Hefin Rhys (Statistician, The Francis Crick Institute) these analyses have been performed. Details of the analyses performed are contained within the Statistics Report submitted and the revised Methods.

Referee #2 (Remarks for Author):

This is a technically sound study that shows that LIMD1, an adaptor protein involved in the down-regulation of HIF transcription factors is regulated by hypoxia and is a direct transcriptional target of HIFs. This suggests a negative feedback loop in hypoxic signaling via upregulation of LIMD1. Moreover hypoxia-induced expression of LIMD1 reduces tumor growth of xenotransplanted tumor cells, and LIMD1 correlates with a better prognosis of lung tumor cases. These are interesting findings which expand previous studies from the group and support a role of LIMD1 in hypoxic signaling in cancer. However there are some specific questions/problems that need to be clarified before publication.

1) Fig. 1 E: It seems more meaningful to show data of Fig. S1G here instead of the deletion analysis because these make the point more clearly by showing specific point mutants of HREs.

We agree with the reviewer and have replaced Figure 1E with Figure S1G.

2) Fig. 2D Why is LIMD1 not induced under hypoxia in this experiment?

*We apologise for the inclusion of this non-representative western blot, which was made in error. We have replaced this with a blot representative of the quantification across biological repeats (**Figure L5A**). We have also included better representative blots for Figure S2B and S2C (**Figure L5B-C**).*

Fig. 2D. Western blot analysis of the indicated proteins in U2OS

Fig. S2B HeLa and

Fig. S2C A549 cells transfected with the indicated siRNA and maintained in normoxia or exposed to hypoxia for 24 hours.

3) Fig. 3B-D needs better description and interpretation: There is a strong drop in HIF1α from 4 hours to 24 hours of hypoxia which is not commented by the authors. Only a minor portion of this drop can be related to exogenous HRE-driven LIMD1 expression because HIF1α is still much lower in the HREmut cells at 24 hrs than at 4 hrs.

The strong drop in HIF-1α protein observed between 4 and 24 hours in cells (in this assay, represented by HRE-WT cell lines) is well characterised in the literature; under conditions of chronic hypoxia, a negative regulatory feedback loop is initiated whereby free oxygen from inhibited mitochondrial respiration leads to over-activation of PHDs, causing HIF-α degradation and a desensitised hypoxic response (17). This is now clarified in the manuscript (Page 3, lines 12-15):

‘Under conditions of chronic hypoxia, a negative regulatory feedback loop is initiated whereby free oxygen from inhibited mitochondrial respiration leads to over-activation of PHDs, causing HIF-α degradation and a desensitised hypoxic response.’

In the HRE-MUT cell lines, hypoxic induction of LIMD1 expression is impaired; this reduces the efficiency of HIF- α degradation under conditions of chronic hypoxia. However, it is important to note that our manuscript does not claim that LIMD1 is the sole factor regulating the adaptive response to hypoxia. Indeed we have previously published that degradation of HIF-1 α occurs in both a LIMD1-dependent and -independent manner (18). LIMD1 does not have any enzymatic activity, instead it acts as a molecular scaffold for the PHD-VHL axis enhancing HIF degradation. Therefore other factors, such as PHD2/3 upregulation, will still be active in mediating HIF degradation in the absence of LIMD1, albeit at lower efficiency. Hence in the HRE-MUT lines, we observe a reduction in efficiency of hypoxic HIF-1 α turnover, as opposed to a complete absence of this adaptive mechanism.

In line the differences in HIF reporter activity and target gene expression between HERwt and HERmut cells are relatively small, raising the question of their significance. The authors should address this issue and also use shorter time points to capture the optimal time window for LIMD1 activity.

The small but statistically significant differences we observe in our HRE-WT/HRE-MUT isogenic paired cell lines are not entirely surprising: LIMD1 is still expressed under hypoxic conditions in the HRE-MUT line, but these cells do not benefit from enhanced hypoxic induction of LIMD1 from a functional HRE (Figure 3C-E). HRE-WT cells exhibit a 2-3 fold increase in LIMD1 expression under hypoxic conditions, which does not occur in the HRE-MUT lines. Following the reviewers advice, we repeated the experiment in the U2OS, HeLa and primary non-transformed SAEC isogenic cell line pairs, with additional time-points of hypoxic incubation to now include 0, 4, 8, 16, 24 and 48 hours (Figure 3F-G) with appropriate amendments to figure legends, methods and text (Page 8, lines 8-14):

‘U2OS, HeLa and SAEC were transduced with the paired set of lentiviruses described. We identified within this non-clonal population, LIMD1 controlled by the HRE^{wt} promoter had a 2-3 fold hypoxic-induction of protein expression (Fig. 3B-E and Fig. S3B-E). In contrast, mutation of the HRE within the LIMD1 promoter (HRE^{mut}) significantly impaired hypoxic induction of LIMD1 in these lines. This was coupled with an impairment of HIF-1 α degradation under increasing exposure to hypoxia in the HRE^{mut} lines compared to HRE^{wt} (Fig. 3B, F-H).’

However, as we still see maximal hypoxic induction of LIMD1 at 24 hours in HeLa and U2OS paired lines, we still believe this to be the optimum time point for the reporter assay and target gene experiments to be performed. As our manuscript shows, small effects from modulating LIMD1 expression in vitro result in large and profound changes in vivo (Figure 5, 6, S4-7), therefore we believe the subtle differences observed, to be significant and of biological importance.

Is the massive decline in HIF1 α between 4 and 24 hrs due to endogenous-LIMD1 which is not efficiently depleted by the shLIMD1?

Regarding the presence of endogenous-LIMD1 affecting HIF degradation, we do not believe this would be sufficient to cause any significant effects. The reasoning briefly for this is demonstrated by lentiviral transduction of U2OS cells with the shRNA vector from which the HRE-WT and HRE-MUT lines were generated, (the parental vector targets LIMD1 but does not contain the rescue component), which shows highly efficient (>95% estimated) knock-down of LIMD1 (Response Figure 3A). This also demonstrates that the rescue component of the shRNA vector re-expresses LIMD1 from its endogenous promoter at levels close/equivalent to endogenous (as observed in the shSCR line).

Response Figure 3: These data are not included in the manuscript

(A) U2OS cells transduced with the indicated lentiviral vectors and analysed by western blot for the indicated proteins demonstrate highly efficient LIMD1 knock-down by the parental shLIMD1 vector.

4) Why are differences between flag bands from HREwt and HREmut much smaller than between LIMD1 bands if both blots detect the same (exogenous) protein? Again, what is the contribution of endogenous LIMD1 in this setting and how well did the shRNA to LIMD1 perform in these cells?

The affinity of the anti-Flag antibody for its epitope is greater than that of the endogenous LIMD1 antibody, hence the differences in the observed intensities in the original figure; although it is important to note that densitometric analysis of these blots demonstrated the same significant difference in expression at 24 and 48 hour time points. However, we have since re-optimised the blotting conditions of the Flag antibody and include western blots that are representative for both LIMD1 and Flag.

5) Related to the above, in Fig. S3B it appears that there is no specific increase of LIMD1 in the HREwt transduced cells under hypoxia, rather there seems to be a decrease in the HREmut transduced cells. How can the authors explain this?

*As suggested by the reviewer for the U2OS HRE cell lines, we have repeated this experiment in the HeLa HRE isogenic pair with additional time-points of hypoxic incubation to now include 0, 4, 8, 16, 24 and 48 hours (**Figure 3B-H, Figure S3B-E**). We observe a 3-fold induction of LIMD1 in the HRE^{wt} line by 24 hours hypoxic incubation, which is significantly impaired in the HRE^{mut} line, LIMD1 levels are maintained (**Figure 3D and S3D**). The adaptive response to hypoxia is impaired in the HRE^{mut} line, where HIF-1 α levels are increased compared to their HRE^{wt} counterpart (**Figure 3G**). We apologise for inclusion of a non-representative western blot for this cell line pair in the original version.*

6) Fig. 7 Please explicitly state whether correlations between LIMD1 and HIF1 or LiMD1 and VEGF expression are positive or negative. It is not possible to extract this information from Fig. S7.

*We refer Reviewer 2 to Point 6 (above Reviewer 1) where we have addressed this point. Briefly, we have determined that VEGF was not a robust hypoxic-marker to interrogate, particularly for a small lung adenocarcinoma cohort. We have therefore performed bioinformatic analysis using TCGA data and identified that HIF target genes SLC2A1, GAPDH and IGF2BP2 are significantly inversely correlated with LIMD1 expression in the lung adenocarcinoma cohort (**Figure 6F-H**). Kaplan-Meier analysis of these genes determined high expression was correlated with shorter overall survival (**Figure 6I-K**). These have now been used in place of VEGF.*

7) The discussion section might be shortened; in particular with respect to parts dealing with VEGF targeted therapies where the connection to the presented data is rather loose.

We agree with the reviewer and have removed the indicated section.

3B

3C

3D

3F

3F

3G

3H

Fig. 3B. U2OS, HeLa and SAEC isogenic HRE^{WT} and HRE^{MUT} pairs were exposed for hypoxia for the indicated time-points and analysed for expression of Flag-LIMD1 (RNAi resistant form) and HIF-1 α .

Fig. 3C-H. Western blot analysis LIMD1 (C-E) and HIF-1 α (F-H) in U2OS, HeLa, and SAEC isogenic pairs in (Fig. 3B), normalised to β -Actin and 0 hour hypoxic time-point for each cell line.

Figure S3B-E. Densitometric analysis of LIMD1 protein expression for U2OS (S3B-C), HeLa (S3D) and SAEC (S3E) isogenic cell line pairs, mean \pm SEM, n=3.

2. Gazdar AF, Gao B, Minna JD. Lung cancer cell lines: Useless artifacts or invaluable tools for medical science? *Lung cancer* (Amsterdam, Netherlands). 2010;68(3):309-18.
3. Ertel A, Verghese A, Byers SW, Ochs M, Tozeren A. Pathway-specific differences between tumor cell lines and normal and tumor tissue cells. *Molecular cancer*. 2006;5(1):55.
4. Pientka FK, Hu J, Schindler SG, Brix B, Thiel A, Jöhren O, et al. Oxygen sensing by the prolyl-4-hydroxylase PHD2 within the nuclear compartment and the influence of compartmentalisation on HIF-1 signalling. *Journal of cell science*. 2012;125(Pt 21):5168-76.
5. Rantanen K, Pursiheimo J, Hogel H, Himanen V, Metzen E, Jaakkola PM. Prolyl hydroxylase PHD3 activates oxygen-dependent protein aggregation. *Molecular biology of the cell*. 2008;19(5):2231-40.
6. Steinhoff A, Pientka FK, Mockel S, Kettelhake A, Hartmann E, Kohler M, et al. Cellular oxygen sensing: Importins and exportins are mediators of intracellular localisation of prolyl-4-hydroxylases PHD1 and PHD2. *Biochemical and biophysical research communications*. 2009;387(4):705-11.
7. Hergovich A, Lisztwan J, Barry R, Ballschmieter P, Krek W. Regulation of microtubule stability by the von Hippel-Lindau tumour suppressor protein pVHL. *Nature cell biology*. 2003;5(1):64-70.
8. Schraml P, Hergovich A, Hatz F, Amin MB, Lim SD, Krek W, et al. Relevance of nuclear and cytoplasmic von hippel lindau protein expression for renal carcinoma progression. *The American journal of pathology*. 2003;163(3):1013-20.
9. Sharp TV, Al-Attar A, Foxler DE, Ding L, de AVTQ, Zhang Y, et al. The chromosome 3p21.3-encoded gene, LIMD1, is a critical tumor suppressor involved in human lung cancer development. *Proceedings of the National Academy of Sciences of the United States of America*. 2008;105(50):19932-7.
10. Sharp TV, Munoz F, Bourbouli D, Presneau N, Darai E, Wang HW, et al. LIM domains-containing protein 1 (LIMD1), a tumor suppressor encoded at chromosome 3p21.3, binds pRB and represses E2F-driven transcription. *Proceedings of the National Academy of Sciences of the United States of America*. 2004;101(47):16531-6.
11. Tsai WB, Long Y, Chang JT, Savaraj N, Feun LG, Jung M, et al. Chromatin remodeling system p300-HDAC2-Sin3A is involved in Arginine Starvation-Induced HIF-1alpha Degradation at the ASS1 promoter for ASS1 Derepression. *Scientific reports*. 2017;7(1):10814.
12. Spendlove I, Al-Attar A, Watherstone O, Webb TM, Ellis IO, Longmore GD, et al. Differential subcellular localisation of the tumour suppressor protein LIMD1 in breast cancer correlates with patient survival. *International journal of cancer*. 2008;123(10):2247-53.
13. Ghosh S, Ghosh A, Maiti GP, Mukherjee N, Dutta S, Roy A, et al. LIMD1 is more frequently altered than RB1 in head and neck squamous cell carcinoma: clinical and prognostic implications. *Molecular cancer*. 2010;9:58-.
14. Pinato DJ, Ramachandran R, Toussi ST, Vergine M, Ngo N, Sharma R, et al. Immunohistochemical markers of the hypoxic response can identify malignancy in pheochromocytomas and paragangliomas and optimize the detection of tumours with VHL germline mutations. *British journal of cancer*. 2013;108(2):429-37.
15. Raleigh JA, Calkins-Adams DP, Rinker LH, Ballenger CA, Weissler MC, Fowler WC, Jr., et al. Hypoxia and vascular endothelial growth factor expression in human squamous cell carcinomas using pimonidazole as a hypoxia marker. *Cancer research*. 1998;58(17):3765-8.
16. Wykoff CC, Beasley NJ, Watson PH, Turner KJ, Pastorek J, Sibtain A, et al. Hypoxia-inducible expression of tumor-associated carbonic anhydrases. *Cancer research*. 2000;60(24):7075-83.
17. Ginouves A, Ilc K, Macias N, Pouyssegur J, Berra E. PHDs overactivation during chronic hypoxia "desensitizes" HIFalpha and protects cells from necrosis. *Proceedings of the National Academy of Sciences of the United States of America*. 2008;105(12):4745-50.
18. Foxler DE, Bridge KS, James V, Webb TM, Mee M, Wong SC, et al. The LIMD1 protein bridges an association between the prolyl hydroxylases and VHL to repress HIF-1 activity. *Nature cell biology*. 2012;14(2):201-8.

Thank you for the submission of your revised manuscript to EMBO Molecular Medicine. We have now received the enclosed reports from the referees that were asked to re-assess it. As you will see

the reviewers are now supportive and I am pleased to inform you that we will be able to accept your manuscript pending final editorial amendments.

Please submit your revised manuscript within two weeks. To accelerate the process, make sure to provide a point-by-point response to my letter.

I look forward to seeing a revised form of your manuscript as soon as possible.

***** Reviewer's comments *****

Referee #1 (Remarks for Author):

The authors have adequately responded to most of the primary review points, and have added additional analyses. Despite some reservations regarding micro vessel immunodetection, the manuscript is suitable for publication.

Referee #2 (Comments on Novelty/Model System for Author):

The authors have dealt well with the critical aspects raised previously and the ms has improved considerably.

Referee #2 (Remarks for Author):

It is a bit puzzling that in two figure panels "non-representative" Western blots had been submitted initially. Nevertheless, the authors have dealt well with the critical aspects raised previously and the ms has improved considerably.

2nd Revision - authors' response

23 May 2018

We have now addressed all the issue you raised and now hope these are all in order and that our manuscript can now be accepted for publication.

Corresponding Author Name: Tyson V. Sharp

Manuscript Number: EMM-2017-08304-V2